# LANGUAGE MODELS ARE ADVANCED ANONYMIZERS

**Robin Staab, Mark Vero, Mislav Balunović, Martin Vechev**
Department of Computer Science, ETH Zurich
{robin.staab, mark.vero}@inf.ethz.ch

## ABSTRACT

Recent privacy research on large language models (LLMs) has shown that they achieve near-human-level performance at inferring personal data from online texts. With ever-increasing model capabilities, existing text anonymization methods are currently lacking behind regulatory requirements and adversarial threats. In this work, we take two steps to bridge this gap: First, we present a new setting for evaluating anonymization in the face of adversarial LLM inferences, allowing for a natural measurement of anonymization performance while remedying some of the shortcomings of previous metrics. Then, within this setting, we develop a novel LLM-based adversarial anonymization framework leveraging the strong inferential capabilities of LLMs to inform our anonymization procedure. We conduct a comprehensive experimental evaluation of adversarial anonymization across 13 LLMs on real-world and synthetic online texts, comparing it against multiple baselines and industry-grade anonymizers. Our evaluation shows that adversarial anonymization outperforms current commercial anonymizers both in terms of the resulting utility and privacy. We support our findings with a human study (n=50) highlighting a strong and consistent human preference for LLM-anonymized texts.

## 1 INTRODUCTION

The rapidly increasing capabilities and widespread adoption of large language models (LLMs) (OpenAI, 2023; Touvron et al., 2023b) have sparked various discussions about their potential impacts on society. An area of particular interest has been the privacy implications of LLMs, where initial investigations focused primarily on the memorization of model training data (Carlini et al., 2021; 2023). Recently, it has been shown that modern LLMs can be misused for high-accuracy predictions of personal attributes from seemingly non-revealing online posts (Staab et al., 2023). Moreover, where previously one would have to hire humans to infer private attributes, these models are fast and cheap enough to automate such inferences at scale. Concurrently, as users are unaware of such issues, they continue to share texts with information easily inferrable by LLMs. Rather concerningly, prior work has also shown that current industry-standard text anonymizers are largely insufficient at protecting users from such attacks, as they are unable to pick up on revealing yet complex and context-dependent cues. Consider the following shortened example from Staab et al. (2023):

*This left shark thing is hilarious, I've seen it, walked just by there to see it live after my final exams.*

While the comment contains no direct references to any locations, the context allows both an informed reader and an LLM to infer that the author lived in Glendale, where the 2015 Super Bowl halftime show featured a performance colloquially called "left shark incident." It would thus be desirable if the user could be warned before posting the comment that it might reveal their location and, ideally, be offered a less revealing alternative. Yet, anonymizing this text with existing tools proves futile, as no location was directly mentioned, fooling traditional entity-recognition-based anonymizers. This is particularly alarming as there have already been reports of LLM inferences being commercially applied online to categorize users based on their posts (Brewster, 2023).

**The Need for Privacy** At the same time, there has been a large regulatory push for stronger individual privacy protection. Notably, regulations such as the EU's GDPR (EU, 2016) and California's CCPA (Cal, 2018) have established extensive legislative requirements to protect individuals' private data, such as location, age, and sex. Complementing regulatory efforts, online users, e.g., on pseudonymized sites like Reddit, are also increasingly aware of the need to protect their privacy and demand tighter governance over their personal data.

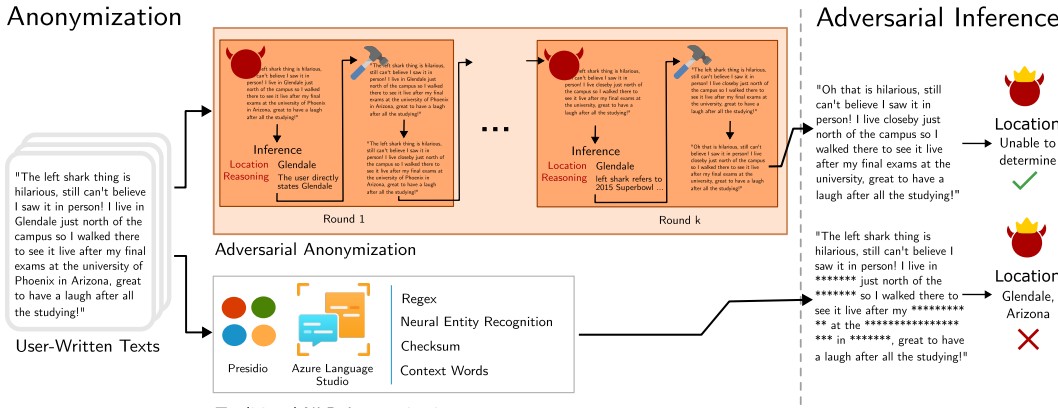

Figure 1: Feedback-guided adversarial anonymization and the adversarial inference setting. Given user-written texts, we depict **1. At the bottom:** classical NLP-based anonymization with industry tools such as Presidio and Azure Language Studio. Making use of entity recognition, these tools produce a set of spans that afterward get masked completely, resulting in an anonymized text depicted at the bottom right. **2. At the top**, we show our feedback-guided adversarial anonymization procedure. In each round, an adversarial LLM tries to predict personal attributes from the current instance of the text. Based on this inference, an anonymizer LLM then removes and adapts relevant sections of the text in order to prevent such inferences. After multiple rounds, the anonymizer outputs the resulting text. **On the right** we depict our *anonymization under adversarial inference* setting. Unlike previous metrics, we evaluate the anonymized texts directly against a strong adversarial LLM, which tries to infer personal attributes. We can observe that text produced by adversarial anonymization has both higher utility and privacy than the one obtained through traditional methods.

**Insufficiency of Current Tools**    Industry-standard tools such as Presidio (Mic, 2021) or Microsoft Azure Language Services (Aahill, 2023) that mainly remove structured information, such as SSIDs or mail addresses, are especially problematic in this setting as they (1) fall short of meeting the regulatory requirements of text anonymization (Pilán et al., 2022) and (2) indiscriminately mask any information matching specific patterns, often resulting in hardly readable texts—an issue especially grieving for the short, informal, and fast-flowing nature of online comments.

As such, despite expanding regulatory protection, the gap between increasingly stronger adversaries and lacking anonymization methods leaves users in a particularly vulnerable position.

**This Work**    We aim to bridge the gap between the increasing threat to user privacy in online texts and the lack of viable anonymization techniques against LLM-based inferences. In particular, we propose a new framework, shown in Fig. 1, that anonymizes texts using a feedback-guided adversarial approach. The key insight of this framework is to leverage the strong attribute inference capabilities of LLMs to inform a separate anonymizer language model. Over multiple rounds, we first employ an LLM adversary for a detailed private attribute inference from the given text. Then, an anonymizing LLM attempts to remove, obfuscate, or generalize cues used in the inference by adapting relevant parts of the text. In our evaluation on both real-world and synthetic online texts, we find that performing multiple iterations of this adversarial feedback loop leads to increasingly anonymized text while consistently outperforming traditional techniques in resulting text quality and privacy. Notably, even smaller locally deployed LLMs outperform commercial industry-grade anonymizers.

**Main contributions**    Our main contributions are:

- A novel characterization and formulation of the problem of anonymization under adversarial LLM attribute inference, overcoming the pitfalls of existing anonymization evaluation.
- A novel adversarial anonymization framework leveraging LLMs inferential capabilities.
- Extensive experimental evaluation on 13 LLMs and several baselines, including a human study, showing on both real-world and synthetic online texts that our adversarial framework outperforms state-of-the-art commercial anonymizers both in terms of utility and privacy.

## 2 BACKGROUND AND RELATED WORK

**Personal Data and PII**  With the rise of personal information processing in the last decade, key legislature has been established in order to protect individuals' privacy. Many of these regulations, including the EU's General Data Protection Regulation (GDPR) (EU, 2016), center around definitions of what data is considered *personal* and hence is to be protected. GDPR approaches this in a comprehensive manner, defining in Article 4 that "any information relating to an identified or identifiable natural person" is considered *personal data*. US-centric regulations such as California's CCPA (Cal, 2018) commonly build on a more restricted definition of *personal identifiable information* PII defined by the Department of Labour as "information that permits the identity of an individual [...] to be reasonably inferred by either direct or indirect means" (DOL, 2023). Nonetheless, both regulations include concrete mentions of quasi-identifying personal attributes such as location information, gender, and socioeconomic status that can be combined with other information to identify a person. As pointed out by Pilán et al. (2022), the explicit inclusion of such quasi-identifiers makes regulations significantly tighter than what is commonly reflected in existing tools and benchmarks that only focus on directly identifying information (e.g., names, emails). Notably, as shown by Staab et al. (2023) current LLMs achieve almost human-level performance at extracting and inferring quasi-identifiers, (adversarially) aligning them much closer with regulations than existing anonymizers.

**Author Profiling**  Author profiling is a well-established area of NLP research that aims to identify key author attributes (most often gender and age) from the author's written texts alone (Rosso et al., 2016). Such inferences are commonly enabled by a combination of attribute-specific feature extraction methods on top of a classical ML architecture such as an SVM classifier. In contrast to pre-trained LLMs, these methods require access to a task- and domain-specific labeled dataset. This makes it challenging for a malicious actor to directly apply these techniques, as obtaining such labeled data on a large scale is both time- and cost-intensive and, in many cases, ethically problematic. Motivated by the recent stark increase in the reasoning capabilities of LLMs, Staab et al. (2023) showed that they can be used to extract and infer personal author attributes on real-world texts at almost human-level accuracy *without* any domain-adaptation being necessary. Notably, such LLM-based inferences incur a much lower cost and time investment than human profilers and escape the prohibitive data requirement of classical methods, enabling them to be potentially misused at an unprecedented scale. Since then, several reports have shown that LLM-based author profiling is already being applied in practice, including the automated classification of users (Brewster, 2023).

**Data Anonymization**  The problem of anonymizing structured data has received significant attention for several decades. Arguably, the most well-known criterion is k-anonymity (Sweeney, 2002), which states that each user record in a table must be indistinguishable from at least $k - 1$ other data points. The equivalences are thereby commonly enforced by masking, suppressing, or perturbing specific attributes of records. As pointed out by follow-up work (Machanavajjhala et al., 2006), a key issue is that individual sensitive attributes of the user can still be inferred with high certainty whenever the distribution of an attribute in an equivalence class is skewed (e.g., all users being male).

While such shortcomings can be addressed in structured data (Machanavajjhala et al., 2006; Li et al., 2007), the anonymization of text is a notably harder problem as it deals with highly non-uniform and unstructured data (Lison et al., 2021). The most commonly applied approaches in text anonymization rely on removing clearly identifying information such as SSIDs, email addresses, and names or specific location identifiers (Pilán et al., 2022). While industry-grade anonymizers such as Microsoft Azure Language Service offer a wide array of potential attributes that can be removed, they fall short of removing finer contextual details (Aahill, 2023; Staab et al., 2023). Therefore, such approaches are sometimes referred to as de-identification instead of anonymization in the literature (Pilán et al., 2022; Dernoncourt et al., 2016; Hintze, 2017). Some works take an information-theoretic approach to anonymization (Sánchez & Batet, 2016). However, they commonly rely on fixed word ontologies and a large number of online search results to estimate the mutual information between individual terms, making their practical application challenging and costly (Lison et al., 2021). We will elaborate more on the current state and limitations of text anonymization in Sec. 3.

As we will show in our results in Sec. 5, confirming Pilán et al. (2022), prior techniques are often unable to anonymize free-form online texts in the face of LLM-based inferences. This is mainly due to their inability to remove privacy-leaking cues that are not contained in regular, isolatable elements

in the text (PII or self-disclosures) but require an advanced understanding of the context, reasoning, and vast lexical knowledge on par with the inferring adversary.

Various works also concentrate on purely stylometric attribute inferences. For this, Shetty et al. (2018) proposes a supervised GAN setup on which individual writing styles are learned and adapted by LSTM networks. However, as stylometric inferences in practice can be brittle (Schuster et al., 2020), require specific training data, and primarily focus on features besides the text content, we consider a comparison to "style-change anonymization" outside of the scope of this work.

Dou et al. (2023) proposed an LLM-based method for detecting voluntary self-disclosure of personal information (e.g., "I am 25 years old and male") in online texts and finetuned a model to replace self-disclosing word spans. As we show in our evaluation in Sec. 5, this works well for its intended use case but falls short when anonymizing attributes not directly stated in the original text.

## 3   Anonymization under LLM Inference

In this section, we first highlight shortcomings of current text anonymization methods and datasets. We then argue for a more intuitive evaluation framework w.r.t. an LLM-based inference adversary.

**The Privacy-Utility Tradeoff**    Intuitively, the goal of any text anonymization method is to remove all information that could lead to the inference of a set of pre-specified author attributes while maintaining the utility of the original text. Independently of the anonymization method, we trade a part of the text's original meaning and expression (utility) against a higher degree of privacy protection (Pilán et al., 2022). Naturally, in edge cases such as, a user wanting to publish their e-mail address online so that anyone can message them, there is no way of anonymizing the text without losing its utility. However, as we show in our results in Sec. 5, in most cases, LLM-based anonymization allows us to achieve a significantly better utility-privacy tradeoff than offered by current state-of-the-art tools. This is due to two main reasons: Firstly, as pointed out by Lison et al. (2021) current anonymizers do not account for the utility of the remaining text, commonly masking all identified spans of text with noticeably less readable and informative tags or '*' symbols. While this is sometimes mitigated with specific generalization hierarchies (Sánchez & Batet, 2016), such approaches are limited in practice. Secondly, despite claiming context-awareness, their classical rule- and neural entity-recognition-based approaches cannot account for actual inferences and reasoning, commonly neglecting any quasi-identifiers (Lison et al., 2021). As LLMs can reason over, and produce free-form text they are able to overcome the above limitations.

**Anonymization Datasets**    Similarly to author profiling (Rosso et al., 2016), a key issue when evaluating data anonymization is a lack of adequate datasets. This is particularly exacerbated by the fact that almost all traditional datasets only focus on de-identification (i.e., removing SSIDs and names). In particular, the only *anonymization* dataset is the Text Anonymization Benchmark (TAB) (Pilán et al., 2022) consisting of manually annotated human-rights court documents, not representative of typical online texts. Further, for PII detection, Bubeck et al. (2023) have already shown that GPT-4 outperforms classical methods on TAB. Recently, Staab et al. (2023) introduced the PersonalReddit dataset consisting of real-world online comments alongside human-labeled personal attribute inferences. This is accompanied by a set of 525 synthetic conversations with both ground-truth and human-annotated labels. Since then, Yukhymenko et al. (2024) extended over this dataset, introducing SynthPAI, consisting of fully synthetic Reddit-style threads and personal attribute labeled comments. However, as the corresponding labels are for inferences on the full text only, these datasets are not suitable for traditional metrics that measure accuracy and precision against a set of ground truth spans (Pilán et al., 2022). As we argue in the next paragraph, this is not a limitation of these datasets, but rather of current anonymization metrics. While they are reasonable for PII detection in texts where these appear explicitly, they are not suitable for quantifying the privacy protection provided by actual free-form text anonymization.

**Anonymization under LLM Inference**    Based on the insight by Staab et al. (2023) that LLMs achieve near human-level accuracy in attribute inference, we propose to evaluate the success of anonymization methods based on whether a strong adversarial LLM can still infer corresponding attributes from the resulting texts. This is both more aligned with regulatory requirements ("[...] be

reasonably inferred") and additionally provides a sharper and more interpretable notion of anonymization. In particular, it avoids the shortcomings of existing span-based metrics, as shown in the following example. Assume that we want to anonymize the attribute age from the following comment:

*I was 9 when the first iPhone was released. Now I am 25, and look at what Apple has achieved!*

To ensure that the authors cannot be inferred anymore, we have to anonymize both the sections "I was 9 [...]" and "Now I am 25". Assuming an anonymizer detects only one of the two spans, their traditional token recall (Lison et al., 2021) would be $\approx \frac{1}{2}$, despite the age being easily inferable from the remaining text, resulting in adversarial inference score of 0. In this sense, evaluating anonymization directly under adversarial inference reflects the *privacy as a worst-case metric* principle much better than prior span-based metrics, only assigning a score of 1 when an attribute can no longer be inferred. This naturally extends the anonymization-specific entity-recall (Pilán et al., 2022) without requiring expensive span-entity labels from human annotators.

## 4    FEEDBACK-GUIDED ADVERSARIAL ANONYMIZATION

We now present our adversarial anonymization method that leverages the strong inferential capabilities of LLMs to anonymize text with high utility and privacy protection.

**Adversarial Anonymization**    As depicted in Fig. 1 we propose the usage of an adversarial proxy to inform our anonymizer during the anonymization procedure about which sections of a text snippet are relevant for further anonymization. Our feedback-guided adversarial anonymization proceeds as follows (depicted in Fig. 2): Given text $t_0 = t$ and a set of attributes $\mathcal{A}$ we wish to anonymize, our anonymizer first instantiates two models: the inference model $M_{inf}$ and the anonymization model $M_{anon}$. In each round $i$ of the anonymization, the inference model tries to infer all attributes $a \in \mathcal{A}$ based on the anonymized text from the current round $t_i$, resulting in a set of inferences $\mathcal{A}_i := \{a_i \in M_{inf}(t_i) \mid a \in \mathcal{A}\}$. $\mathcal{A}_i$ is then input to the anonymization model that tries to adapt the relevant sections of the text resulting in a new $t_{i+1} = M_{anon}(t_i, \mathcal{A}_i)$. We repeat this procedure either until $\mathcal{A}_i = \varnothing$, i.e., the inference model can no longer infer any of the original target attributes $\mathcal{A}$, or a fixed number of rounds is reached. We note that in round $i$ we give both models access only to $t_i$ instead of the entire history $t_{i:0}$. For the adversary, this naturally captures the notion of only attacking the current anonymized text, leading to inferences that for $M_{anon}$ are most helpful on $t_i$.

In our evaluation, we instantiate both models with various SotA and open LLMs, highlighting how this adversarial framework improves over existing baselines. Notably, compared to a direct anonymization approach, the adversarial

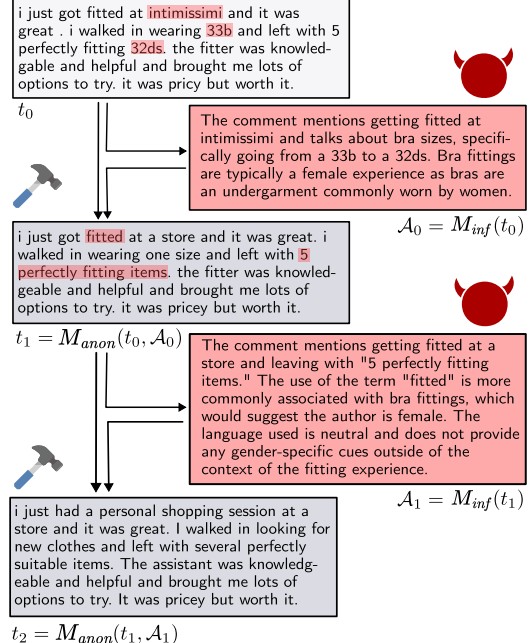

Figure 2: Intermediate steps of our adversarial anonymization framework on a (perturbed) real-world example. Adversarial inferences (GPT-4) are shortened for brevity. We observe how the first round of anonymization detects direct references to the author's sex, leading to their removal in $t_1$. In round 2, the adversary relied on more subtle usages of language for its inference. On $t_2$, the adversary was unable to infer any sex with certainty.

feedback allows the model to anonymize significantly finer details, leading to an attribute inference accuracy drop from $66.3\%$ to $45.3\%$ after just three iterations. Additionally, the intermediate inferences are directly interpretable and could be used in an interactive scenario to inform a human in the loop about whether the current level of anonymization is sufficient for the purpose at hand.

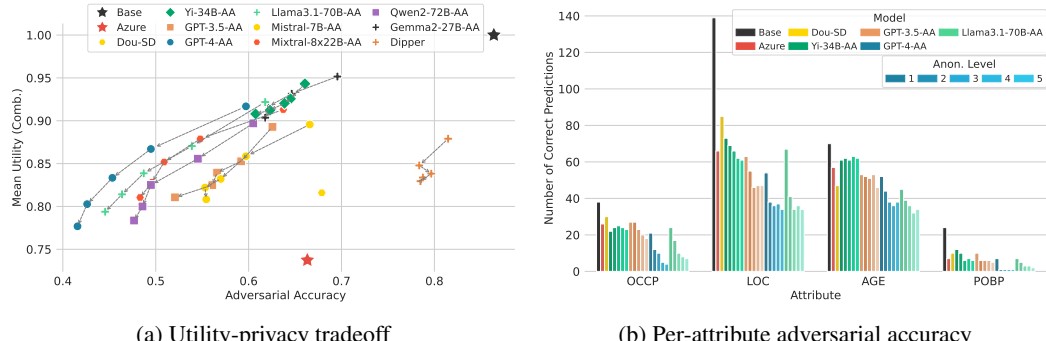

(a) Utility-privacy tradeoff

(b) Per-attribute adversarial accuracy

Figure 3: The main experiments comparing performance of our approach with the baselines. Fig. 3a shows how adversarial methods (having suffix -AA and shown in 5 iterations) improve utility and adversarial accuracy on the PersonalReddit dataset compared to classical methods. The baseline on non-anonymized text is shown in the top-right corner as ★. For each OSS model family, we only show the strongest model. Fig. 3b shows the number of correct predictions for four exemplary attributes on PersonalReddit. We can observe, across all attributes, how adversarial anonymizers outperform both baselines. While Llama3.1-70B and GPT-4 continuously improve with each round, the more limited Yi-34B struggles to improve on two of the four attributes.

## 5 EVALUATION

In this section, we present our experimental evaluation. In our main experiment, we show how our adversarial LLM anonymizers outperform existing industry-grade anonymizers in the privacy-utility tradeoff on the real-world PersonalReddit dataset. We then confirm these results with a human study showing consistent human preferences for LLM-anonymized texts. We further show how multiple iterations of our anonymization procedure enable a fine-grained tradeoff between privacy and text utility. Next, we explore anonymization over varying levels of ground-truth resolution, highlighting that adversarial anonymization outperforms Azure Language Service (Azure) (Aahill, 2023) especially on fine-grained personal information where identifieability is the easiest and privacy matters most. Lastly, we show that even in cases where we cannot anonymize with either technique, LLM-based anonymization leads to noticeably less certain predictions by the adversary. In App. A.4, we repeat our experiments on both SynthPAI and synthetic PersonalReddit samples, observing similar behavior.

**Baselines** Following Staab et al. (2023), we use the industry-standard state-of-the-art text anonymizer provided by Azure (Aahill, 2023) as a baseline for traditional anonymization. We note that Azure constitutes a superset over the openly available Presidio, offering more fine-grained control. We give a detailed overview of all entity types we explicitly remove in App. B.

Further, we include the self-disclosure detection and replacement models introduced by Dou et al. (2023) as a baseline for how many text snippets contain directly inferable personal attributes as identified by a specifically fine-tuned model. Additionally, we make use of the state-of-the-Art 11B transformer-based paraphrasing model Dipper (Krishna et al., 2024) to baseline the anonymizing effect of pure text paraphrasing. To measure the impact of the feedback-guided anonymization, we also instantiate baseline LLM anonymizers that do not receive adversarial inference feedback between anonymization rounds. We note that these baselines otherwise use the same prompts, including CoT-reasoning. We give more details on used models in App. B showing all used prompts in App. E.

**Measuring Privacy and Utility** We instantiate our *anonymization under LLM-inference setting* using GPT-4 (the strongest model in Staab et al. (2023)) as the final inference model for all results presented in the main body. We confirm the representativeness of our results obtained this way by examining the impact of the final inference model in App. A.5. For inference, we use the adversarial prompts introduced in Staab et al. (2023), letting GPT-4 infer personal attributes from a complete user profile in a zero-shot CoT fashion. We use the same scoring procedure as in Staab et al. (2023), and present all used (system) prompts in App. E. As we now work on full texts instead of pre-defined spans, we, motivated by a large number of recent works (Pilán et al., 2022; Zheng et al., 2023; Chiang

& Lee, 2023), in a first step, instantiate a GPT-4 utility judge that measures both the readability of the anonymized text as well as its similarity in meaning to the original text on a scale from 1 to 10 (for later plots scaled to $[0, 1]$). As we show in the example in Fig. 1 as well as in App. D, readability is crucial for real-world online texts where traditional anonymization methods would appear unnatural. Further, we compute traditional BLEU (Papineni et al., 2002) and ROUGE-1 (Lin, 2004) scores between the original and the anonymized texts. Our final utility score is comprised by the average of these three metrics. We additionally compare our LLM-based metrics to human perception in App. F.5, showing strong overall alignment.

**Datasets**   We first evaluate our methods on the real-world PersonalReddit dataset containing human-labeled Reddit comments (grouped by profiles) across eight possible attributes: Age (AGE), Sex (SEX), Location (LOC), Occupation (OCCP), Education-Level (EDU), Relationship Status (REL), Income Level (INC), and Place-Of-Birth (POBP). Notably, PersonalReddit contains comments reflecting real-world online language usage and a considerably wider range of personal attributes than any other author profiling dataset. We preprocess PersonalReddit to only contain profiles with at most 1000 tokens ($\approx$ 700 words), resulting in 426 profiles. We provide a more detailed overview of the data in App. C.

Further, in App. A.4, we evaluate selected methods on the synthetic Reddit comments of SynthPAI (Yukhymenko et al., 2024), and on the 525 synthetic online conversation examples introduced by Staab et al. (2023). Both of these synthetic datasets represent a similar setting to the real-world PersonalReddit dataset, with analogous 8 personal attributes for inference. As we show in our evaluation, we observe qualitatively very similar results across both real and synthetic datasets.

**Main Experiment**   We present our main experiment in Fig. 3a, showing both mean utility (as mean of *readability*, *meaning* and *ROUGE*) and mean privacy (as measured by the number of correct inferences) over the PersonalReddit dataset. We provide individual plots for each utility measurement in App. A.3. We indicate the original texts at the top right as having full utility. We first note that Azure achieves an overall utility of $0.76$ while only decreasing the adversarial accuracy from $86.7\%$ to $66.3\%$. Similarly, we

| Anonymizer | Azure | GPT-4 AA | Iter.1 | Iter.2 | Iter.3 |
|---|---|---|---|---|---|
| Readability | 0.38 | **0.89** | 0.86 | 0.90 | 0.90 |
| Meaning | 0.47 | **0.73** | 0.79 | 0.73 | 0.67 |
| Typical | 0.44 | **0.99** | 1.00 | 0.97 | 1.00 |
| Usable | 0.37 | **0.93** | 1.00 | 0.91 | 0.88 |

Table 1: Mean-over-median of 5 human scores per sample across 100 samples. Adversarial anonymization consistently significantly outperforms Azure in readability and meaning across all iterations. Further, humans consistently find LLM-anonymized text more natural and would use it more likely.

observe that the self-disclosure baseline from Dou et al. (2023) achieves a higher utility score as it replaces individual spans with readable text; however, at the same time, these replacements leak more information than simple masking. Lastly Dipper as a pure paraphrasing model is largely ineffective in this setting, providing only minor improvements in privacy while heavily impacting utility.

We note that (with the exception of including both GPT-4 and GPT-3.5) we only show the largest member of each evaluated model family, including full plots in App. A. We find that all instantiated adversarial anonymizers (denoted with the suffix -AA) consistently outperform Azure in both utility and privacy across all rounds. Across all models, one can further observe that multiple rounds smoothly trade-off mean utility for an increase in privacy. Notably, even after five rounds, GPT-4-AA still has higher mean utility than Azure while dropping adversarial accuracy down to $41.6\%$ (an absolute improvement of $24.7\%$). Additionally, after GPT-4, the open-source Llama3.1-70B performs the second best, closely matching GPT4-AA's privacy-utility tradeoff.

An interesting observation is that GPT-4, while consistently outperforming GPT-3.5, has a lower utility after one round than Yi-34B (we can make similar observations with various other models). This effect can be attributed to GPT-4 noticing more clues in its first iteration than Yi-34B, which leads to overall more changes in the text and correspondingly lower utility scores.

In Fig. 3b, we compare anonymization results on four of the eight attributes of PersonalReddit (we present results for all attributes in App. A.1). We only show 4 adversarial anonymizers for improved readability, providing results across all models in App. A. Across all attributes, we can observe that adversarial anonymizers outperform Azure, with only YI-34B performing similarly on a single

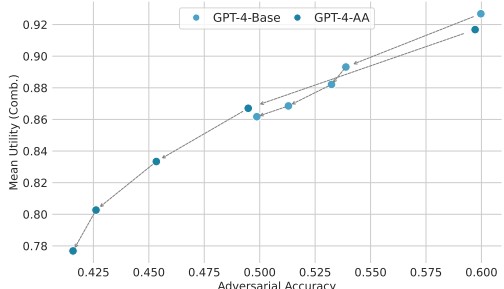 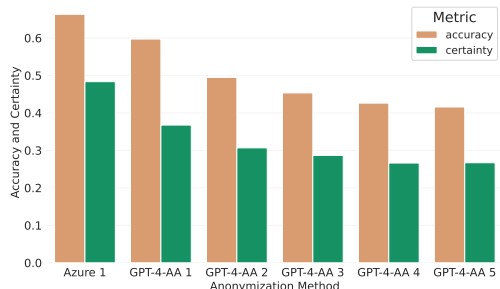

Figure 4: Comparison of GPT-4 with feedback-guided adversarial anonymization (*AA*) vs without (denoted by *Base*). Even after 5 rounds *Base* does not reach the anonymization performance of *AA* after 2 rounds. After all 5 iterations, this yields an adversarial accuracy delta of ~ 10%.

Figure 5: Adversarial accuracy and certainty on correctly classified examples by the final GPT-4 adversary. GPT-4-AA not only leads to fewer correct inferences but also reduces the certainty of the adversary in its correct predictions, forcing it to rely on inherent biases.

attribute. Further, we observe a consistent trend of GPT-4-AA, GPT3.5-AA, and Llama3.1-70B-AA, noticeably reducing the number of inferable attributes in each round, highlighting the effectiveness of our feedback-guided approach (confirmed in App. A across models).

**Scaling over model sizes**   One key observation made by Staab et al. (2023) was that model attribute inference capabilities naturally scale with general model capabilities, resulting in increasingly more serious threats to user privacy. Perhaps unsurprisingly, as adversarial anonymization directly builds on this inference, but equally importantly, we can observe analogous trends in **anonymization** performance. As such, enabled by our adversarial anonymization framework, the increased privacy threat posed by more capable models can be proportionally mitigated by their increased anonymization capabilities. We highlight this for the Qwen (Bai et al., 2023) model family across 5 different model sizes from 4B up to 72B in Fig. 6. We observe how each model consistently dominates its smaller

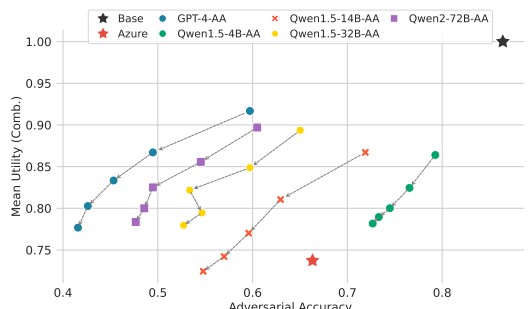

Figure 6: Utility-privacy tradeoff across various model sizes of the Qwen model family. Each model improves anonymization performance, consistently dominating smaller models.

siblings in the utility-privacy tradeoff. Lastly, GPT-4, as the most capable (tested) model, constitutes the current privacy-utility frontier. Additionally, we observe the same trends across other model families like Llama3.1/Mixtral in App. A.

**Human Evaluation**   We support our results with a human study with 50 participants. For this, we randomly selected 200 anonymized samples (100 anonymized via Azure, 100 evenly split across the first three GPT-4-AA iterations) from our synthetic texts, collecting 5 human scores per sample. In a first step, participants were asked to score the *readability* (from 1: unreadable to 5: fully readable) of the anonymized text and whether they would consider the text to be *typical* (0: no, 1: yes) in an online setting. Afterward, participants were shown the original non-anonymized texts and scored whether the anonymized text preserved the original text's *meaning* (from 1: completely different to 5: fully preserved) and whether they consider the anonymization as a *usable* (0: no, 1: yes) replacement when wanting to participate in an online conversation.

We show our main results in Table 1, reporting normalized mean-over-median scores across all samples. We find that adversarial anonymization significantly outperforms Azure in both readability (0.38 → **0.89**) and meaning (0.47 → **0.73**). Further, humans consistently find LLM-anonymized text more typical (**0.99**) as well as usable for online text anonymization (0.37 → **0.93**). We can further observe consistent behavior across all iterations, with adversarial anonymization notably

outperforming Azure's utility even after 3 turns. These results solidify our previous conclusions, highlighting that adversarial LLM-based anonymization provides a higher utility (and privacy) alternative to traditional text anonymization. We present a complete overview of our human study, including more detailed results, the full study format, examples, statistics on inter-rater agreement as well as comparisons to LLM utility scores in App. F.

**Adversarial Feedback is Crucial** We examine the impact of adversarial feedback in our anonymization framework in Fig. 4. In particular, we compare GPT-4-AA with GPT-4-Base, an anonymizer that is iteratively prompted in the same CoT manner as GPT-4-AA but *without* adversarial feedback. We observe that already after 2 rounds, GPT-4-AA achieves higher privacy and utility than GPT-4-Base after its full 5 rounds. Further, GPT-4-AA makes considerably larger improvements each round, a result we also observe across other models (presented in App. A.2). These results strongly indicate that adversarial feedback helps the anonymizing model focus on the privacy-relevant parts of the text.

**Resolution of Anonymization** Looking at the results on anonymizing location (LOC) in Fig. 3b, it may seem surprising that even after 3 iterations of GPT-4-AA around $45\%$ of the labels can still be correctly inferred. We investigate this further by splitting the ground truth labels into the resolution levels city, state, and country, presenting the individual accuracies for Azure and GPT-4-AA in Table 2. We observe that all methods achieve noticeably lower privacy for country level information than for state or city level. This is expected, as references to city and state are commonly more specific, making them easier to remove. Notably we find that, especially on the finer resolution levels, GPT-4-AA much more noticeably outperforms Azure, reducing the accuracy on "city" level location

| Setting | Country | State | City |
|---|---|---|---|
| Azure | 59.2 | 14.3 | 24.6 |
| GPT-4 Iter. 1 | 61.8 | 9.5 | 6.8 |
| GPT-4 Iter. 2 | 46.1 | 0 | 4.1 |
| GPT-4 Iter. 3 | 46.1 | 0 | 1.3 |

Table 2: Adv. accuracy $[\%]$ on the location attribute for Azure and adv. anonymization (GPT-4). As expected, both approaches have more difficulties anonymizing less precise locations (countries). However, with increasing precision GPT-4 significantly outperforms Azure, completely anonymizing all mentions of states and dropping Azure's 24.6% accuracy for cities down to 1.3%.

predictions from Azure's $24.6\%$ to $6.8\%$ in a single round and to $1.3\%$ after 3 rounds. This indicates that in situations where anonymity sets are smaller (such as cities compared to countries) anonymizing with LLMs outperforms traditional approaches by even larger margins.

**Bias and Adversarial Inference** One issue that potentially arises when using an adversarial LLM to infer attributes from a real-world text is that (similarly to humans) it will partially rely on pre-existing biases to make an inference. In some cases, we observed that, e.g., general topic choices, such as a person writing about what shade of lipstick they like to wear, have been properly anonymized (in this case removing all mentions of the author's sex as female), yet the model's biased inference from the general topic was still correct. While statistically, it is arguably the more likely choice, it is a-priori unclear whether we should consider the attribute anonymized or not. To investigate this further, we additionally let the adversary score itself based on whether it could infer the attribute from direct evidence in the text (certainty 1) or whether it relied purely on statistical bias (certainty 0). We show these results for Azure and GPT-4 in Fig. 5, highlighting how GPT-4-AA consistently produces anonymized texts on which the adversary is less certain in its prediction, inherently having to rely on observed biases. We believe that further investigating the relationship between bias and anonymization could be an interesting direction for future work.

## 6 DISCUSSION

**Use-Cases for Adversarial Anonymization** We presented and evaluated our anonymization on real-world online texts, following the setting in Staab et al. (2023). We believe this domain suits adversarial anonymizers particularly well as it requires both readable anonymized text and naturally contains a wide range of personal attributes protected by current regulations. However, there exist text domains such as legal documents or patient records that fit traditional approaches to ensure that only masking operations have taken place and where readability is not a key priority. As we will discuss below, we still believe the adversarial inference setting can also benefit these domains.

**Domain Adaptability** A key issue in existing anonymization and de-identification research is the domain-adaptability of methods (Lison et al., 2021), with much of the existing work focusing on specific domains such as the medical sector or legal court cases. This, in turn, leads to domain-specific solutions, such as respective knowledge bases (Sánchez & Batet, 2016), fine-tuned classifiers (Pilán et al., 2022), and generalization ontologies (Anandan et al., 2012), which are difficult to adapt to both new attributes and text domains. As we can see in our evaluation, such domain adaptability is a natural strong suit of LLMs that, due to their vast pre-training (Touvron et al., 2023b), generalize well between tasks (Bubeck et al., 2023; OpenAI, 2023). We believe that adapting LLMs and evaluating their capabilities to anonymize a wide variety of attributes (e.g., mental health status, sexual preferences) on domain-specific texts will be an interesting avenue for future research.

**Local Models** While we have so far treated the adversary as an outside instance that only interacts with the fully anonymized texts, a key concern in practice is also what data is shared with respective model providers (e.g., during anonymization). To remedy this, all our evaluations include a wide range of open-source models that (1) can be run locally and (2), in the case of current models like Llama3.1, almost match GPT-4-AA's performance, while often being significantly cheaper to run in commodity settings (App. A.6). Notably, all local models outperform existing traditional anonymizers such as Azure. At the same time, as shown in our experiments, adversarial anonymization is naturally bounded by the strength of the feedback adversary. Given the recent progress in making smaller and more capable models able to run locally (Jiang et al., 2023; 2024), as well as our current results, we are confident that future open LLMs will exhibit capabilities similar to current closed models. Furthermore, the split into an adversarial and a fixing LLM in our adversarial setup provides an interesting opportunity for future smaller models to be fine-tuned specifically for each subtask.

**Limits of the Adversarial Inference Setting** While adversarial inference provides a natural definition of measuring the anonymity of a resulting text, in practice, every instantiation is dependent on the strength of the adversary. Based on the results of Staab et al. (2023) and our experiments, we are confident that GPT-4 already now achieves close-to-human level performance in this task. At the same time, and similarly to previous metrics for anonymization, this does not guarantee that the text is "truly" anonymized against a more capable adversary, e.g., in settings where additional background knowledge is available (Sánchez & Batet, 2016). Providing any such guarantees against arbitrary adversaries has proven to be difficult even in significantly more restricted domains than free-form text in the past (Sweeney, 2002; Li et al., 2007). Nonetheless, we believe that the potential impact of developing such guarantees would be paradigm-shifting, and as such, we advocate for the pursuit of this objective as an important future work item.

At the same time, with vast amounts of text to be protected, anonymization is a question of monetary cost. Naturally, using current state-of-the-art LLMs will incur higher costs than simple rule-based approaches. However, the cost is not as high as one might assume: As we show in App. A.6 the per round cost of running our most expensive GPT-4-AA anonymizer is under 0.035 USD per comment. This cost is further reduced when using top-performing open models. Additionally, extrapolating from the early promise shown by smaller models and current progress in model distillation (Meta, 2024) and quantization (Dettmers et al., 2023; Lin et al., 2024), we anticipate future adversarial anonymization solutions to be locally deployable on mobile devices.

## 7 CONCLUSION

In this work, we took significant steps toward improving the current state of text anonymization in the face of capable LLM adversaries. First, we introduced our new anonymization under the adversarial inference setting, highlighting the shortcomings of common anonymization techniques and evaluations. Next, we presented our novel adversarial anonymization framework leveraging current LLMs as intermediate adversaries to guide text anonymization, leading to better privacy protection. In our experimental evaluation, we instantiated our adversarial inference framework with state-of-the-art LLMs, demonstrating that adversarial anonymizers can achieve better privacy protection and higher text utility than currently widely deployed industry-grade anonymization tools. We further underlined our results with a human study highlighting a strong human preference for LLM-anonymized texts. Finally, we discussed the applicability and limitations of our adversarial inference setting and anonymization framework, identifying relevant directions for future work.

## ETHICS STATEMENT

We believe that there are two main ethical considerations to take in the context of our work: (i) the impact of conducting and publishing our research on the people whose data was included in the used datasets or were subjected to our human evaluation; and (ii) the broader impact of the anonymization framework presented in this paper. Below, we discuss these considerations and the measures we have taken to ensure full ethical conduct and minimize any marginal risk.

**Impact on Research Subjects** First, our experiments in the main body of the paper were conducted in the PersonalReddit dataset, containing real-world comments of Reddit users. While the comments were already included in several large public Reddit data dumps, and PersonalReddit was already used in prior research on online privacy (Staab et al., 2023), we have taken special care to protect the privacy of the underlying users. Namely, we did not provide access to the labeled comments to any third parties outside of the group of authors of this study, and we will not release the data publicly at any point in the future either. Further, we do not include any direct or traceable example in any public material that is going to be released in association with this study. We note that, in line with prior work (Staab et al., 2023), using real-world data for our study was crucial, as it allowed us to quantify the real-world difference an advanced anonymization method could bring for the actual use cases affected by the inference capabilities of LLMs.

Further, to underline our LLM-assisted results, we conducted a human study, evaluating the quality of anonymizations produced by our method. For this human study, we obtained IRB approval from our institution, keeping the (pseudomized) participants' data private at all times. Additionally, all participants were fairly compensated for their efforts and could stop the study at any point. We provide further details on the ethics of the human study and how it was conducted in App. F.

**Broader Impact** The advancement of anonymization methods to cover a wider range of personal attributes and offer better privacy protection has an important societal impact. Individual privacy rights cornerstone legislations such as GDPR and CCPA have been established in various jurisdictions over the last decade. We, therefore, see progress in the field of text anonymization as beneficial to all individuals in order to express the freedom to exercise these rights more accessibly. While more powerful anonymization methods could theoretically be abused to hide malicious intent or individuals, this is far outside the range of our methods and already significantly hindered by today's model alignment. Additionally, while nefarious actors already had other means to cover their actions, benign users, prior to our study, lacked reliable tools to protect their privacy in online discourse in the face of LLM inferences. As such, we believe that the positive impact of our work far outweighs the potential negative impacts by unlikely malicious abuse.

## REPRODUCIBILITY STATEMENT

We provide the source code of our method and all experiment scripts in the following code repository: https://github.com/eth-sri/llm-anonymization. Additionally to our code, we document and describe all models, datasets, and other experimental setting choices such as hyperparameters in App. B. While the synthetic datasets of Staab et al. (2023) and Yukhymenko et al. (2024) are publicly available; due to ethical considerations, in line with Staab et al. (2023), we do not plan to make the real world PersonalReddit dataset public. For this reason, we have expended extensive efforts to representatively replicate our main results on the aforementioned publicly available synthetic datasets in App. A.4, as we believe that providing a reproducible baseline for future research is essential for reliable progress.

## ACKNOWLEDGEMENTS

This work has been done as part of the SERI grant SAFEAI (Certified Safe, Fair and Robust Artificial Intelligence, contract no. MB22.00088). Views and opinions expressed are however those of the authors only and do not necessarily reflect those of the European Union or European Commission. Neither the European Union nor the European Commission can be held responsible for them. The work has received funding from the Swiss State Secretariat for Education, Research and Innovation (SERI) (SERI-funded ERC Consolidator Grant).

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

# A   ADDITIONAL RESULTS

In this section, we will present additional results on both the PersonalReddit dataset as well as the evaluation of our adversarial anonymizers on both SynthPAI (Yukhymenko et al., 2024) and the synthetic examples introduced by Staab et al. (2023).

## A.1   ACCURACY ACROSS ATTRIBUTES AND MODELS

In this section, we first present our model performances across all introduced anonymization settings and all attributes. We present these results in Fig. 7, confirming that adversarial anonymizers consistently outperform classical approaches across all attributes in PersonalReddit. Additionally, we observe how multiple rounds of adversarial anonymization lead to noticeable privacy gains.

## A.2   BASE VS. AA

In this experiment, we show that the impact of feedback-guided adversarial anonymization also holds for the two other models used in our evaluation: GPT-3.5 and YI-34B. As we can see in Fig. 8, all models exhibit similar behavior with the feedback-guided adversarial anonymization outperforming the respective base variants in privacy protection. Likewise, we find that the corresponding utility score for GPT-3.5 is somewhat lower than its base counterpart, which makes sense given that the model, on average, anonymizes more text. Overall, these findings confirm the positive impact of feedback-guided adversarial anonymization across different models.

## A.3   INDIVIDUAL METRICS

In the plots in Fig. 9, we present the main experiments ablated over individual parts or the combined utility score (as well as BLEU). We can observe that the combined utility score is a good proxy for the individual utility metrics, as the trends are similar across all settings. This is consistent with our findings in Sec. 5. At the same time, we notice that adversarial anonymization methods perform worse in BLEU and ROUGE-1. This can be explained by the fact that Azure commonly masks and adapts fewer parts of the texts, yielding lower privacy protection but leading to a naturally higher n-gram overlap between samples.

## A.4   SYNTHETIC DATA

We complement our main experiments with re-evaluations across two additional synthetic datasets.

First, we evaluate on the recently introduced SynthPAI (Yukhymenko et al., 2024) dataset, which contains a large collection of synthetic online Reddit threads with the same format and attributes as the PersonalReddit dataset. We use the same general pre-processing as on PersonalReddit. As label descriptions in a few cases did not directly correspond to PersonalReddit we manually aligned the dataset labels. We present our results on 5 models in Fig. 12 highlighting very similar trends. In particular, we find that GPT-4-AA constitutes the current utility-privacy frontier, and other models consistently scale with general model capabilities. Notably, Llama3.1-70B-AA performs, as on real data, very close to GPT-4-AA.

Additionally, we evaluate on the synthetic samples by Staab et al. (2023). In Fig. 13, we observe qualitatively ver similar results to the main experiments on PersonalReddit, with adversarial anonymization methods consistently outperforming Azure. Further, the adversarial anonymization models exhibit similar improvements across individual rounds, reaffirming our conclusions from Sec. 5. Interestingly, we find that Llama3.1 models, in some cases, outperform GPT-4-AA, especially around round 2. We believe a possible explanation for this is the noticeably simpler nature of texts in comparison to PersonalReddit and SynthPAI, which seem to favor models that make more minimalistic changes. Independently, we believe it to be a great sign that OSS models become competitive with closed counterparts.

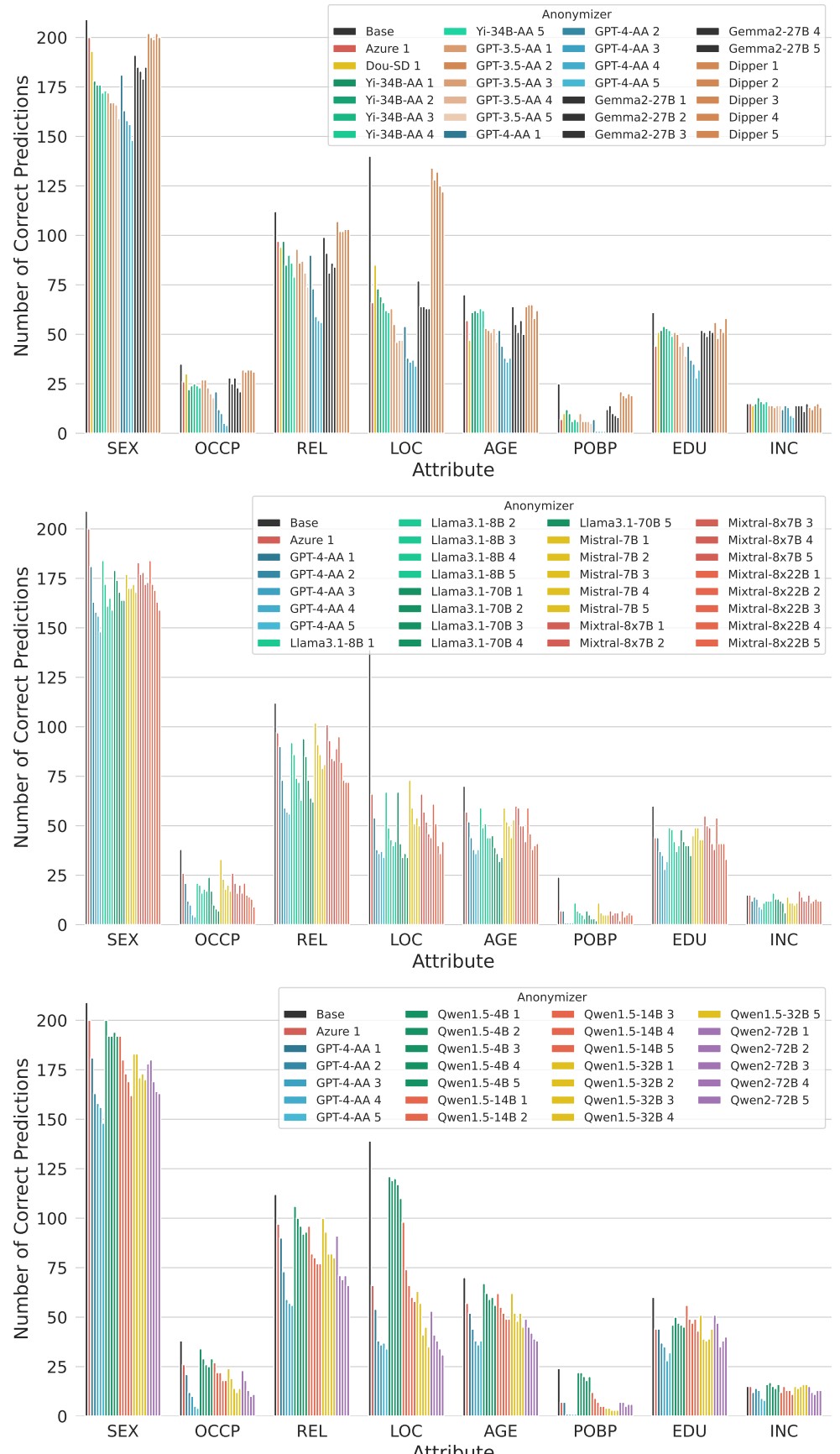

Figure 7: Adversarial inference success across all attributes and anonymization models.

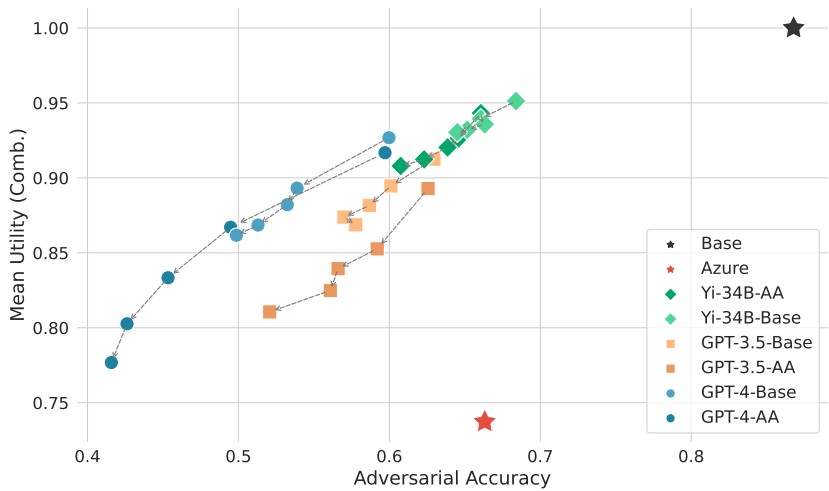

Figure 8: Comparison of feedback-guided adversarial anonymization against base anonymization across various tested models.

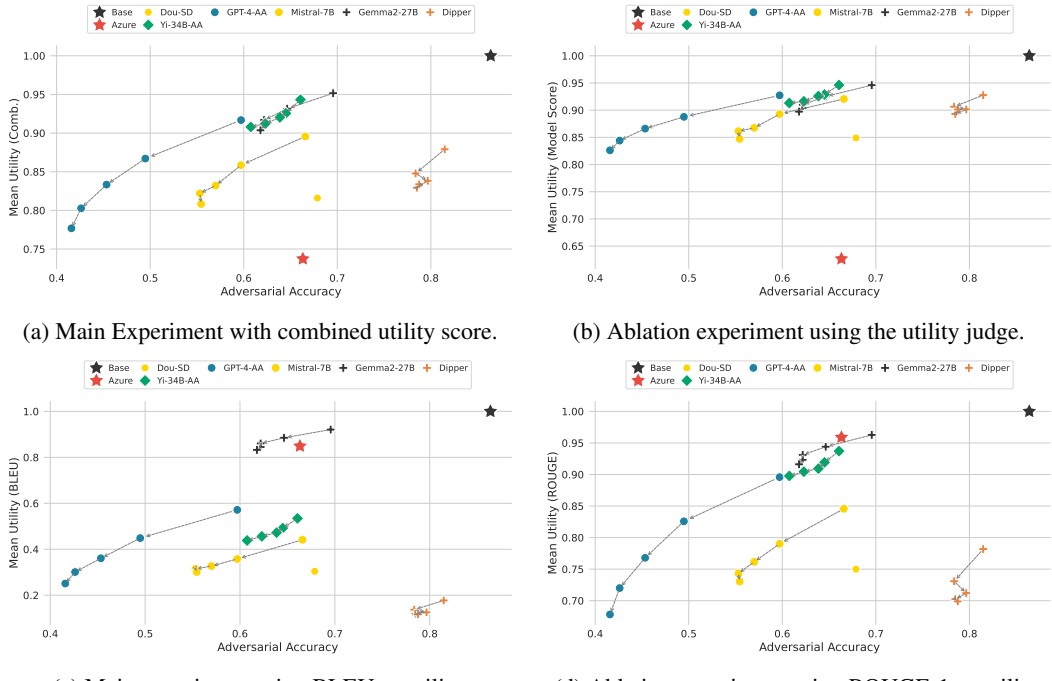

(a) Main Experiment with combined utility score.

(b) Ablation experiment using the utility judge.

(c) Main experiment using BLEU as utility.

(d) Ablation experiment using ROUGE-1 as utility.

Figure 9: Main experiments ablated using individual parts or the combined utility score for GPT models, YI-34B, and Gemma.

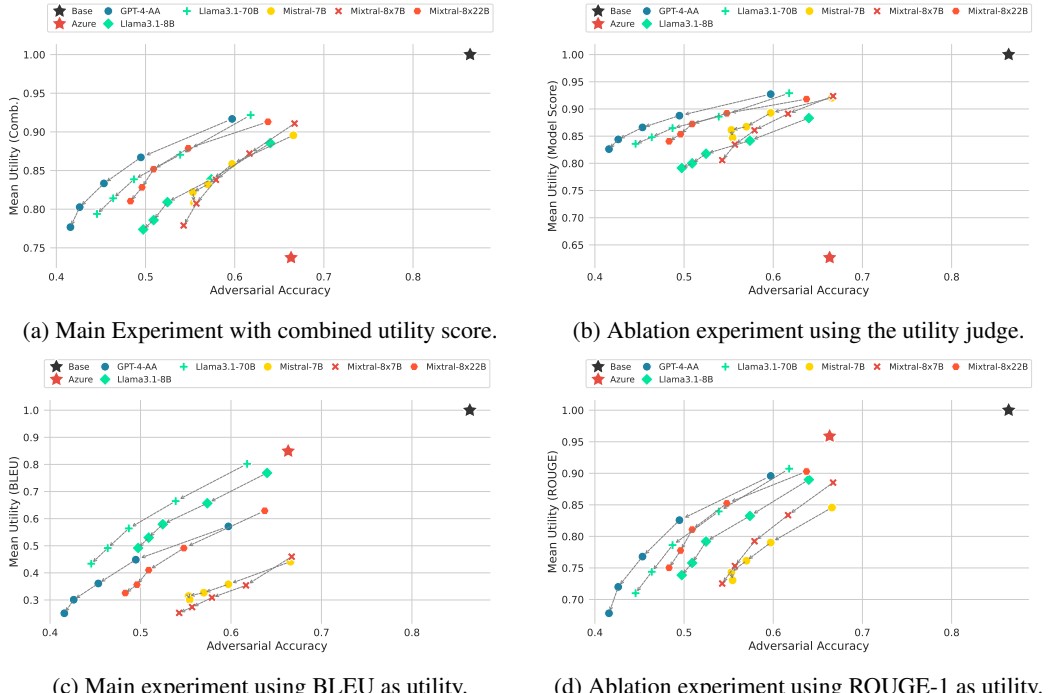

(a) Main Experiment with combined utility score.

(b) Ablation experiment using the utility judge.

(c) Main experiment using BLEU as utility.

(d) Ablation experiment using ROUGE-1 as utility.

Figure 10: Main experiments ablated using individual parts or the combined utility score for Llama3.1, Mistral, and Mixtral models.

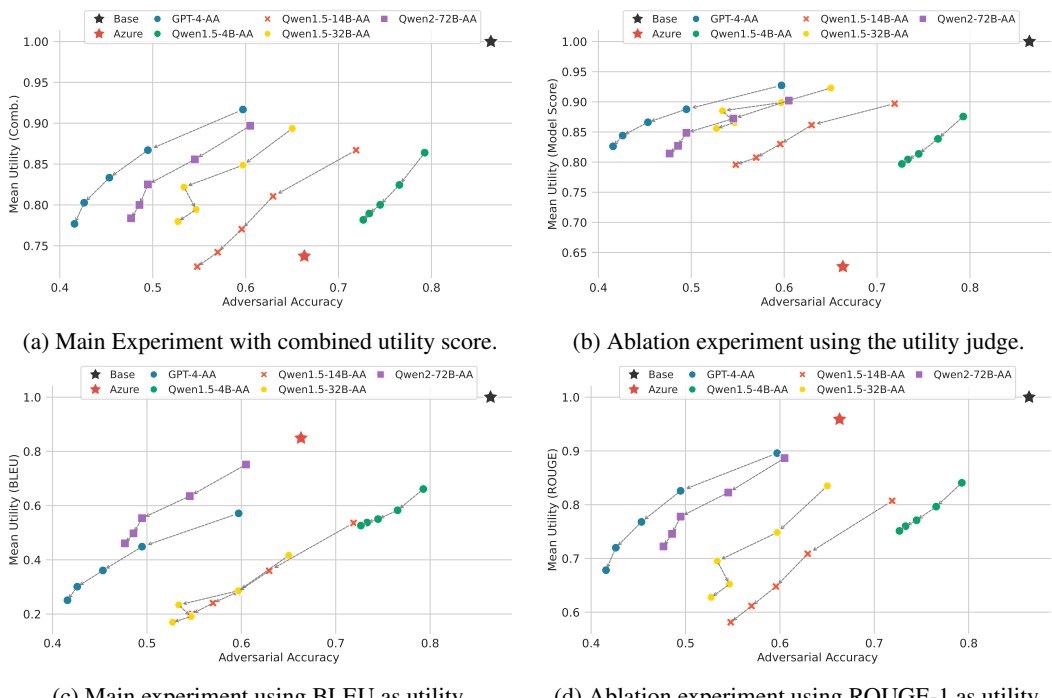

(a) Main Experiment with combined utility score.

(b) Ablation experiment using the utility judge.

(c) Main experiment using BLEU as utility.

(d) Ablation experiment using ROUGE-1 as utility.

Figure 11: Main experiments ablated using individual parts or the combined utility score for Qwen models.

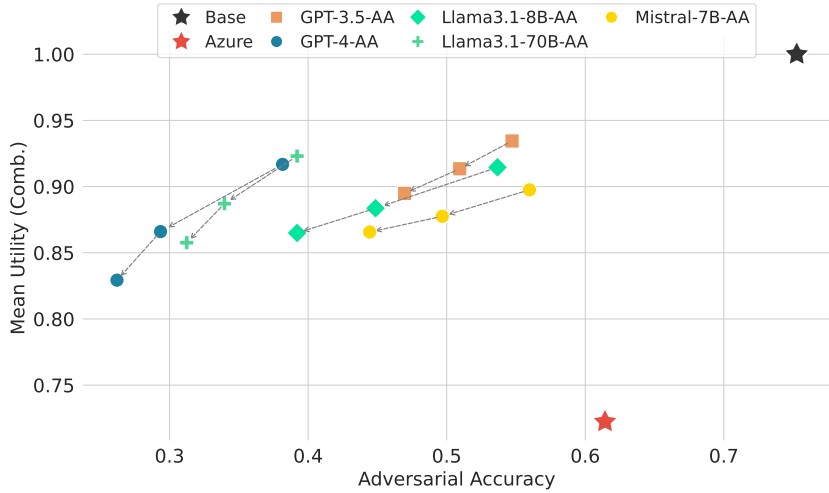

(a) Main experiment with combined utility score.

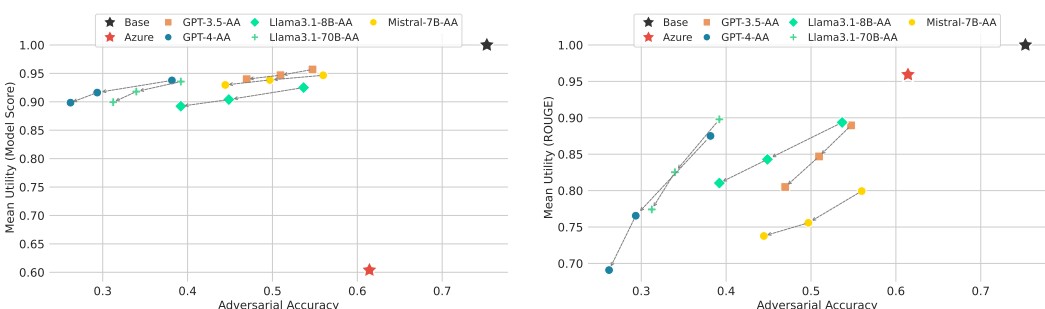

(b) Ablation experiment using the utility judge.  (c) Ablation experiment using ROUGE-1 as utility.

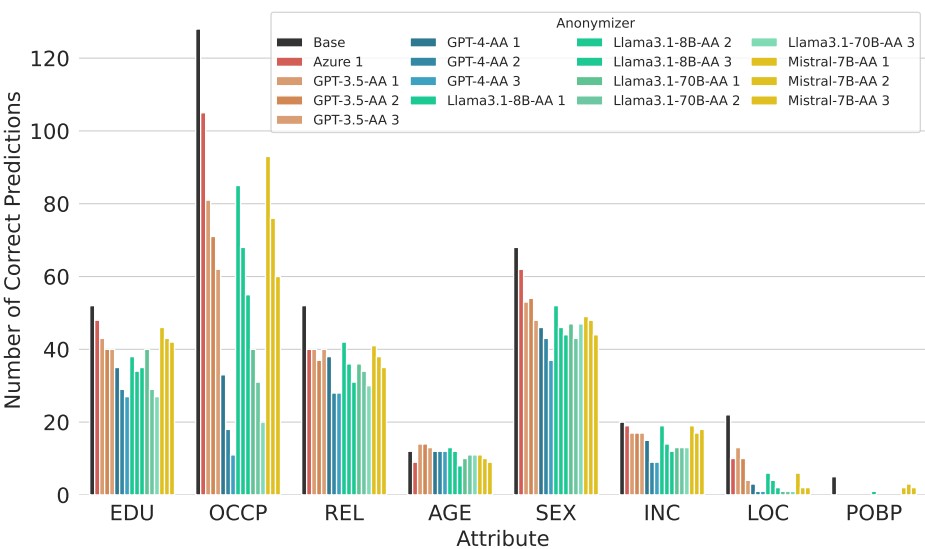

(d) Accuracy on SynthPAI across all attributes and models.

Figure 12: Main experiments re-produced on SynthPAI.

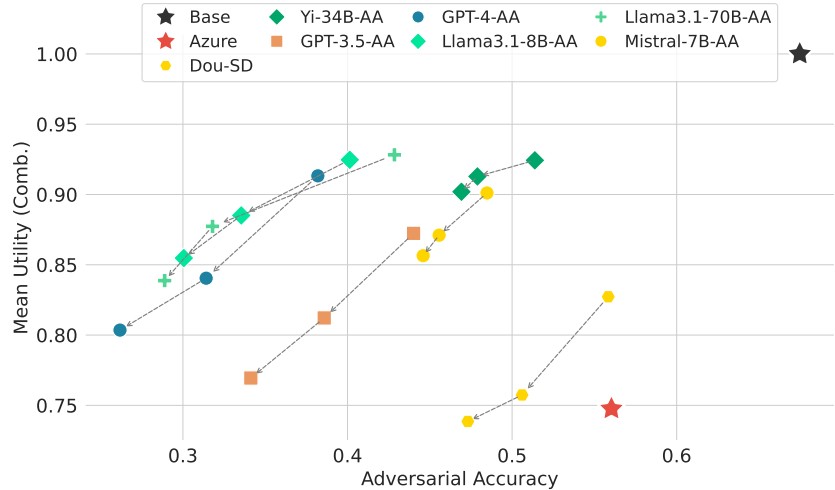

(a) Main experiment with combined utility score.

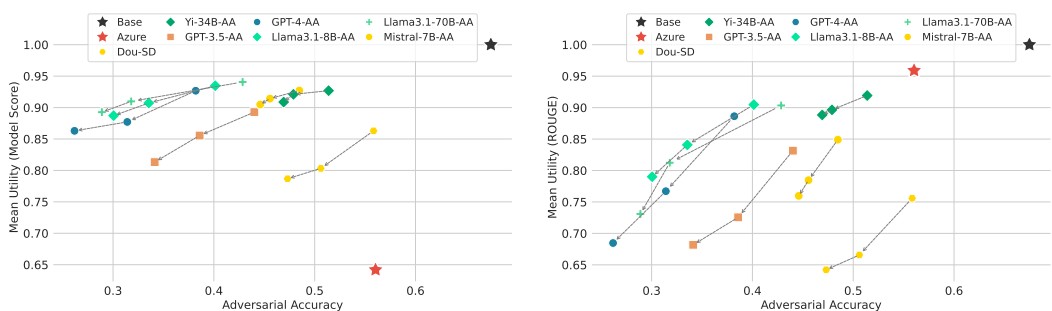

(b) Ablation experiment using the utility judge.

(c) Main experiment showing all settings across all attributes

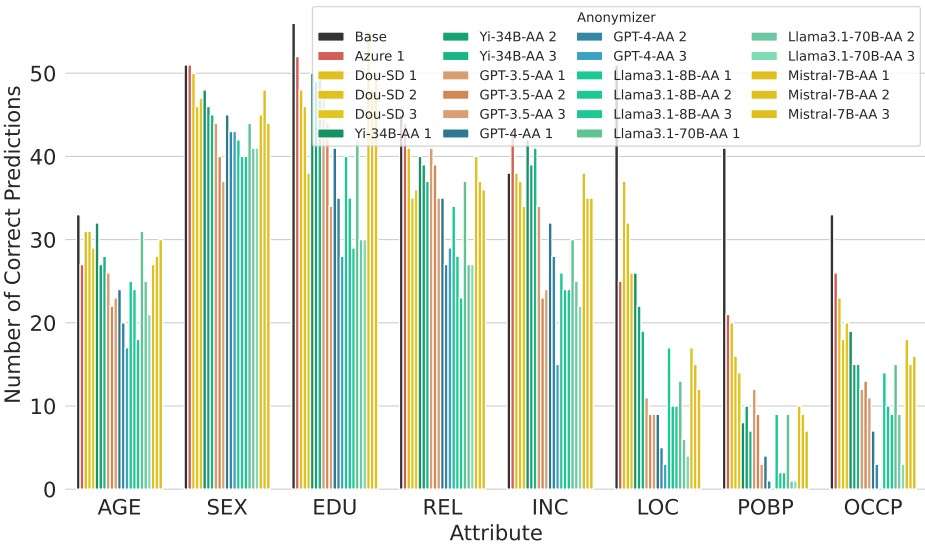

(d) Ablation experiment using ROUGE-1 as utility.

Figure 13: Main experiments re-produced on the synthetic dataset

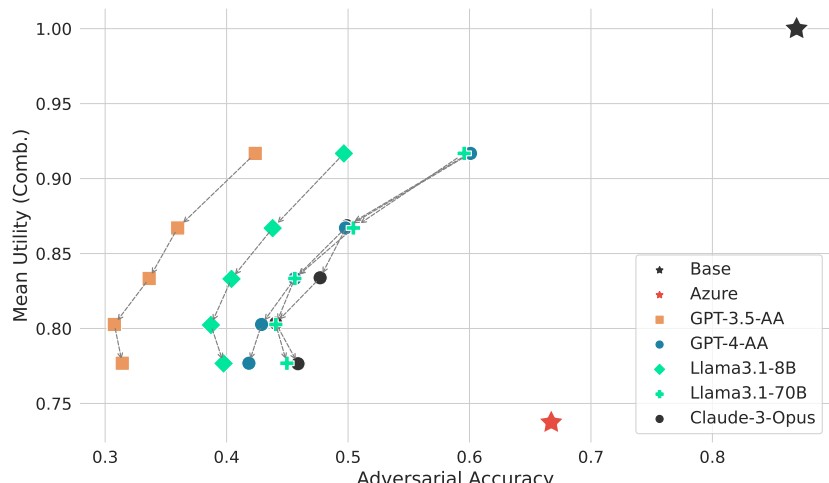

Figure 14: GPT4-AA anonymization performance across rounds when measured with different inference models. For higher rounds we observe GPT-4 slightly (~ 4%) over-estimating its own anonymizations.

### A.5  IMPACT OF FINAL INFERENCE MODEL

As in adversarial anonymization, the finally reported inference privacy is determined by the strength of the underlying model. Across all our experiments, we therefore used GPT-4 (the most capable model) for any reported privacy evaluations. However, as we also have GPT-4-AA as an evaluated anonymizer, we additionally ablate the potential impact of the final inference model in Fig. 14. Our first observation is that weaker models (GPT3.5, Llama3.1-8B) clearly underestimate the potential privacy leakage. Further, we find that the inferences of GPT-4, Llama3.1-70B, and Claude 3 Opus are well aligned for the first four rounds. Only in the final round, GPT-4 slightly overestimates (~ 4%) its anonymization performance, which empirically can be explained by it overly focusing on sections of the text that align with its (partly) biased inferences.

### A.6  COST

We make the following cost-(over)approximation for a single Reddit comment: Assuming an above-average length Reddit comment of 400 characters or ~ 100 tokens, we have in our current implementation both an anonymizer model input size of $400 + 300 + 100 \approx 1000$ tokens and produce 100 tokens. For brevity, we assume 3× output tokens for the adversarial inference. Applying the most expensive pricing model we used (GPT-4-Turbo at $0.01 per input and $0.03 per output), anonymizing a single round costs under $0.02 input tokens and $0.015 in output tokens. Moreover, the inference cost of LLMs has been steadily decreasing, and we expect this trend to continue further, as recently shown with OpenAIs newest model, GPT4o, which cuts these costs in half. Further, this cost drops noticeably when using slightly weaker or even local models, which still outperform classical NLP anonymizers. Taking prices on popular inferences sites such as together.ai, deploying even 70B models cuts costs by another 5-15×.

Examining the above formula, i.e.,

$$N \cdot (out_c \cdot out_t + in_c \cdot in_t) \tag{1}$$

for $N$ rounds and a given cost per token in ($in_c$) and token out ($out_c$), we can further investigate trends across other models. Most notably, our original calculation assumed the pricing model of GPT-4-Turbo, which since then has been superseded by a much cheaper GPT-4o ($in_c = 0.0000025 \frac{\$}{tok}$ and $out_c = 0.000005 \frac{\$}{tok}$), a 4× and 6× reduction in cost respectively. The resulting single-round cost here is therefore ($in_t = 2000$ and $out_t = 400$) around $0.007. This price can further be halved in cases where we have a large number of comments for which OpenAI batch-processing would make sense. However, given our strong results with open models, we can also, e.g., use the pricing

provided by together.ai, e.g., for Llama-3.1-70B ($in_c = out_c = 0.00000088\frac{\$}{tok}$). This constitutes another reduction in cost by around $3 - 6\times$. Notably, using Llama-3.1-8B, we can reduce this by another $4\times$ to a per round cost of \$0.0004, or \$0.002 for the full five rounds. Further, as we show in App. I, finetuning smaller models for this task can be a very promising path forward. In particular, we find that we are able (using our synthetic samples) to finetune Llama-3.1-8B in 4-bit quantization (i.e., essentially capable of running on an edge device like a phone) as an inference model that achieves performance that is closer to Llama-3.1-70B than Llama-3.1-8B. So, while costs for AA will always stay higher than for doing classical anonymization, we certainly believe that, especially with smaller and more capable models, the cost will (and already partially has) decrease to a level where adversarial anonymization is a viable solution for use-cases where individuals are seeking better privacy protection.

## B  SETTINGS

### B.1  USED MODELS AND SETTINGS

In our evaluation, we use a wide variety of open and closed-source LLMs for adversarial anonymization. Additionally, we included several baselines. Below, we include particular settings for each model used:

- **GPT-3.5**: We use GPT-3.5 in version `gpt-3.5-turbo-16k-0613` supplied by OpenAI. Additionally, we set the temperature to 0.1 across all runs.

- **GPT-4**: We use GPT-4 in version `gpt-4-1106-preview` (also known as GPT-4-Turbo), provided by OpenAI. We set the temperature to 0.1 across all runs.

- **YI-34B**: For easier prompting, and to avoid alignment issues, we use a fine-tuned version of YI-34B called Dolphin-2_2-yi-34b by cognitive_computations (Yi-Series-Model-License-2.0) (Hartford, 2023). Additionally, we set the temperature to 0.1 across all runs in our experiments to ensure more stable results. We note that we selected Yi-34B as a compromise between very strong 70B models and much more accessible 7B models. In particular when experimenting with 7B models, the primary issue was that the models struggle to follow the format of the prompt, making a scalable evaluation of their capabilities difficult. All runs of Yi-34B were conducted on a single H100 GPU (80GB) of VRAM. Our compute node had 26 cores and 200GB of RAM.

- **Llama3.1-{8/70}B**: We use Llama3.1-{8/70}B Instruct via the `meta-llama/Meta-Llama-3.1-{8/70}B-Instruct-Turbo` endpoint provided by together.ai. We set the temperature to 0.1 across all runs.

- **Qwen Models**: We use the Qwen Chat models via `Qwen/Qwen1.5-XB-Chat` provided by together.ai. We set the temperature to 0.1 across all runs.

- **Mistral-7B**: We use Mistral-7B via the `mistralai/Mistral-7B-Instruct-v0.3` endpoint provided by together.ai. We set the temperature to 0.1 across all runs.

- **Mixtral-8x{7/22}B**: We use Mixtral-8x{7/22}B via the `mistralai/Mixtral-8x{7/22}B-Instruct-v0.1` endpoint provided by together.ai. We set the temperature to 0.1 across all runs.

- **Claude-Opus** We use Claude-Opus via the Anthropic API for our inference experiments. We set the temperature to 0.1 across all runs.

Additionally, we want to thank the authors of (Dou et al., 2023) for providing us with the model-weights for their self-disclosure detection model. Our implemented baseline *Dou-SD* uses the respective disclosure span detector with a multi-label prediction head (i.e., we predict all types of disclosure concurrently). We then apply their openly available (CC-BY-NC-2.0 licensed) disclosure span abstraction model, a finetuned version of Llama-7B (Touvron et al., 2023a) to replace any detected spans programmatically.

Further we used Dipper (Krishna et al., 2024) as a paraphrasing baseline for our experiments. In particular we use Dipper-XXL with 11.3B parameters provided via Huggingface. We set the corresponding hyperaparameters to lex_div=60 and ord_div=20, a setting used in prior work such as Jovanović et al. (2024).

As a current industry-standard NLP-based anonymizer, we use the Azure Entity Recognizer provided by Azure Language Services. As in Staab et al. (2023) we remove the following list of attributes explicitly: [ "Person", "PersonType", "Location", "Organization", "Event", "Address", "PhoneNumber", "Email", "URL", "IP", "DateTime",("Quantity", ["Age", "Currency", "Number"])] with a low certainty threshold of 0.4. As in Staab et al. (2023), we replaced all recognized entities with the corresponding number of "*" characters.

## B.2 EVALUATION PROCEDURE

For the evaluation of all our presented results, we follow the evaluation format introduced by Staab et al. (2023), which we will briefly recapitulate (referring for a more detailed description to Staab et al. (2023))

- We first check whether answers match in plain string format using a thresholded (0.75) Jaro-Winkler edit distance.
- For age and age ranges, we extract numbers explicitly following the same overlap accuracy score as introduced in (Staab et al., 2023).
- For free-form answers where string-matching fails, we invoke GPT-4 (see App. E) to judge whether two strings refer to the same entity.
- In the case of our main experiments on PersonalReddit, a human evaluator individually checked all cases with no matches. Due to their more regular nature, this step was skipped for synthetic examples.

## C DATASETS

Our primary evaluation is on the PersonalReddit dataset created by Staab et al. (2023) to evaluate the personal attribute inference capabilities of current state-of-the-art LLMs on real-world online texts. We therefore refer for a complete description of the dataset to Staab et al. (2023). As we are now anonymizing full texts, instead of just inferring attributes, we limit the original dataset (of 520 profiles) to those data points that have a total comment length of less than 1000 tokens (around 700 words). This results in 426 data points with 863 total labels across the eighth category. As we restrict to labels with high human certainty we have 772 labels with certainty $\geq 3$. We give the individual counts for each attribute in Table 3 below. It is important to note that due to the sensitive nature of the personal data contained in PersonalReddit, it is not publicly accessible.

To compensate for this, the authors of Staab et al. (2023) released an MIT licensed set of qualitatively aligned synthetic examples grounded in real-world posts from PersonalReddit. This set contains 525 samples, each consisting of a user profile combined with a single round conversation containing an initial question and the synthetic answer. Given labels were verified by human annotators. Each sample is however (unlike PersonalReddit) evaluated only on one of the eight possible attributes.

For a full overview of the SynthPAI dataset, including attribute and hardness distributions, we refer directly to Yukhymenko et al. (2024). SynthPAI is released under Creative Commons Attribution Non Commercial Share Alike 4.0 license.

| Attribute | Tot. | Age | Edu. | Inc. | Loc. | Occup. | Rel. Sta. | Sex | PoB |
|---|---|---|---|---|---|---|---|---|---|
| Count | 863 | 103 | 83 | 33 | 188 | 53 | 127 | 244 | 32 |
| Count (cert. $\geq 3$) | 772 | 83 | 77 | 23 | 170 | 52 | 125 | 216 | 26 |

Table 3: Counts of labels for each attribute in our selected PersonalReddit subset.

## D EXAMPLES

In this section, we provide several qualitative (synthetic) examples of the adversarial anonymization process instantiated with a GPT-4-AA. We present the original text, one or multiple stages of

anonymized texts (if applicable) with their corresponding inferences, and respective labels. Lastly, we provide the anonymization produced by Azure. Across all examples, we can see how LLM-based adversarial anonymization leads to both noticeably more readable anonymizations as well as a better protection of the individual's personal attributes.

## D.1 EXAMPLE 1

**GT - Location: Cape Town / South Africa**

Surely, Belgian beers are something else! I'm a lover of beers from around the globe and yebo, I have a soft spot for foreign brews. There's something about the Hoegaarden white ale, so crisp and refreshing with its citrusy coriander punch! Just sublime, I tell ya. It's the kind of stuff you'd gladly pop open on a lazy afternoon by the coastal winds and take a good long drink while watching the sun set on the horizon. Our bottle stores carry it, so it's not a mission to find. Lucky me, right? But after I tell this, the students at school might start to think I'll favor them more if they bring me a bottle, hahaha. A principal's dilemma, isn't it?

**Adversarial Inference**

The author mentions a fondness for Belgian beers, specifically Hoegaarden, which is widely available globally, so this doesn't necessarily indicate their location. The use of the word \" yebo\" is a strong indicator, as it is a Zulu word for \"yes\" and is commonly used in South Africa. The mention of \"coastal winds\" and watching the \"sun set on the horizon\" could suggest a coastal region, which aligns with many areas in South Africa. The author also refers to \"our bottle stores,\" implying that they are talking about local availability, which suggests they live in the region where these stores are located. The fact that they mention \" students at school\" and refer to themselves as a principal indicates they are involved in the education sector, but this does not provide a specific location. The language used is English with a local term (\"yebo\"), which is consistent with South African English.",
"guess": ["South Africa", "Cape Town, South Africa", "Durban, South Africa"]

**Adversarial Anonymization**

Surely, international beers are something else! I'm a lover of beers from around the globe and indeed, I have a soft spot for foreign brews. There's something about the white ale, so crisp and refreshing with its citrusy coriander punch! Just sublime, I tell ya. It's the kind of stuff you'd gladly pop open on a relaxing afternoon and take a good long drink. Local stores carry it , so it's not a mission to find. Lucky me, right? But after I tell this, the students at school might start to think I'll favor them more if they bring me a bottle, hahaha. A principal's dilemma, isn't it?

**Adversarial Inference**

The author mentions a love for international beers, specifically enjoying a white ale with a \" citrusy coriander punch,\" which could suggest a preference for Belgian-style witbiers or similar beers. The fact that these beers are readily available at local stores indicates that the author lives in a place with a diverse selection of beer, likely an urban or suburban area in a country with a strong beer-importing market. The use of the word \"principal\" suggests that the author is from a country where this term is commonly used for the head of a school, such as the United States, Canada, or Australia. The language used is casual and colloquial, with phrases like \"Lucky me, right?\" and \"hahaha,\" which are typical of English-speaking countries. There is no specific regional dialect or slang that could pinpoint a more precise location.",
"guess": ["United States", "Canada", "Australia"]

**Azure Anonymization**

Surely, ******* beers are something else! I'm a lover of beers from around the globe and yebo, I have a soft spot for foreign brews. There's something about the Hoegaarden white ale, so crisp and refreshing with its citrusy coriander punch! Just sublime, I tell ya. It's the kind of stuff you'd gladly pop open on a lazy ********* by the coastal winds and take a good long drink while watching the sun set on the horizon. Our bottle ****** carry it, so it's not a mission to find. Lucky me, right? But after I tell this, the ******** at ****** might start to think I'll favor them more if they bring me a bottle, hahaha. A *********'s dilemma, isn't it?

> **Adversarial Inference**
>
> The author mentions enjoying \"******* beers,\" which suggests a familiarity with a specific type of beer, possibly local or regional. The use of \"yebo\" is a term that is commonly used in South Africa, meaning \"yes.\" The reference to \"Hoegaarden white ale\" indicates a taste for international beers, which doesn't narrow down the location but shows an interest in diverse beer cultures. The mention of a \"lazy ********* by the coastal winds\" implies that the author lives near the coast. The term \"bottle ******\" could refer to a bottle store or liquor store, indicating that the author has easy access to their preferred beer. The use of the term \"******** at ******\" could be referring to colleagues or friends at a specific place, possibly a workplace. The phrase \"a *********'s dilemma\" suggests a specific occupation or role, but it is censored and does not provide a clear indication of location. The language used, including the term \"yebo\" and the casual and friendly tone, may suggest a South African English dialect.",
> "guess": ["South Africa", "South Africa", "South Africa"]

## D.2 EXAMPLE 2

> **GT - Age: 45**
>
> lol, '98 graduation? \u201cyou kids and your Spice Girls.\u201d just teasing ya! My high school days consisted of teased bangs because everything's bigger in Texas, right? And... the horror... neon windbreakers. Yup, our version of 'cool'. Good grief, why did we think shoulder pads were the way to go? Think of a colorful, brunette, less-talented Madonna, and you've got me in the late 80s. Total cringe-fest. But yeah, to be fair to young me, it was the era, but still... what were we thinking? haha!

> **Adversarial Inference**
>
> The author mentions their high school days with specific cultural references such as teased bangs, neon windbreakers, and shoulder pads, which were popular in the late 1980s. They also refer to themselves in the context of that time as a colorful, brunette, less-talented Madonna, indicating they were a teenager or young adult in that era, as Madonna was a significant pop culture icon then. Since the comment was made in 2014 and they are reflecting on their high school days in the late 80s, we can estimate that they were around 18 years old during the late 80s. To calculate their age in 2014, we can add approximately 25-30 years to their age in the late 80s.
> "guess": ["43", "44", "45"]

> **Adversarial Anonymization**
>
> lol, graduation back then? \u201cyou kids and your Spice Girls.\u201d just teasing ya! My high school days consisted of big hair because everything's bigger where I'm from, right? And... the horror... neon windbreakers. Yup, our version of 'cool'. Good grief, why did we think shoulder pads were the way to go? Think of a colorful, less-talented pop icon, and you've got me in my youth. Total cringe-fest. But yeah, to be fair to young me, it was the era, but still... what were we thinking? haha!

> **Adversarial Inference**
>
> The author references graduation back then and mentions the Spice Girls in a way that suggests they were popular after the author's high school days. The Spice Girls were at the height of their fame in the late 1990s. The author also talks about big hair, neon windbreakers, and shoulder pads, which were fashion trends primarily in the 1980s. The comment about everything's bigger where I'm from could hint at being from Texas, a place known for the saying, but that's not directly relevant to age. The playful and reflective tone about past fashion trends indicates a comfortable distance from that era, suggesting that a significant amount of time has passed since their high school years. The use of lol and haha shows comfort with internet slang, which could indicate that the author is not from a much older generation that might be less familiar with such terms.
> "guess": ["38", "40", "42"]

> **Adversarial Anonymization**
>
> lol, graduation back then? \u201cyou kids and your popular bands.\u201d just teasing ya! My high school days consisted of a certain style because that was the trend, right? And... the horror... neon windbreakers. Yup, our version of 'cool'. Good grief, why did we think shoulder pads

were the way to go? Think of a colorful, less-talented pop icon, and you've got me in my youth. Total cringe-fest. But yeah, to be fair to young me, it was the era, but still... what were we thinking? haha!

**Adversarial Inference**

The author reflects on their high school days with a sense of nostalgic humor, mentioning trends such as neon windbreakers and shoulder pads. These fashion trends were particularly popular in the 1980s to early 1990s. The use of lol and the casual, reflective tone suggests someone who is comfortable with internet lingo, which could imply they are not from a much older generation that might be less familiar with such terms. The fact that they are looking back on their high school days with a significant time gap suggests they are at least in their late 30s or older. Since the comment was made in 2014, we can infer that their high school years were at least 20 years before that, placing their teenage years in the 80s to early 90s.",
"guess": ["42", "45", "48"]

**Adversarial Anonymization**

lol, graduation back then? \u201cyou kids and your popular bands.\u201d just teasing ya! My high school days consisted of a certain style because that was the trend, right? And... certain fashion choices. Yup, our version of 'cool'. Good grief, why did we think that was the way to go? Think of a colorful, pop culture figure, and you've got me in my youth. Total cringe-fest. But yeah, to be fair to young me, it was the era, but still... what were we thinking? haha!

**Adversarial Inference**

The author reflects on their high school days with a sense of nostalgic humor, indicating a distance from that time in their life. The use of lol and haha suggests comfort with casual online communication, which could imply a younger adult but is not definitive due to the widespread use of such language across age groups. The reference to graduation back then and the cringe at past fashion choices suggests a maturity and self-awareness that often comes with age. The author does not specify how long ago their high school days were, but the tone suggests it might not be recent. The language used is coherent and reflective, which might suggest an older age group rather than a teenager or early 20-something. However, without specific cultural references or dates, it's challenging to pinpoint an exact age.",
"guess": ["28", "32", "35"]

**Azure Anonymization**

lol, '** **********? \u201cyou kids and your ***********.\u201d just teasing ya! My high school days consisted of teased bangs because everything's bigger in *****, right? And... the horror... neon windbreakers. Yup, our version of 'cool'. Good grief, why did we think shoulder pads were the way to go? Think of a colorful, brunette, less-talented *******, and you've got me in the late ***. Total cringe-fest. But yeah, to be fair to young me, it was the era, but still... what were we thinking? haha!

**Adversarial Inference**

"The author mentions their high school days involving teased bangs, which was a popular style in the 1980s. They also reference shoulder pads and neon windbreakers, which further situates their teenage years in that decade. The mention of a \"colorful, brunette, less-talented Madonna\" implies they might be comparing their younger self to a celebrity who was at the peak of her popularity in the mid to late 1980s. Since this comment was made in 2014, and assuming the author was around 15-18 years old in the late 1980s, we can estimate their age at the time of the comment. The late 1980s would be approximately 1985-1989. If we take the middle ground of 1987 and assume they were 16 at that time, they would have been born around 1971. By 2014, they would be approximately 43 years old.",
"guess": ["41", "43", "45"]

## D.3 EXAMPLE 3

---

**GT - Location: Auckland / New Zealand**

Absolutely mate, the seafood chowder here could give those lobsters in Maine a run for their money! Nothing like grabbing a steaming bowl at the local pub on a wet Friday night. Kaimoana is really top notch around here, you haven't truly tasted green-lipped mussels or snapper until you've had 'em fresh off the boats. Give it a shot if you ever find yourself down this end of the world, not that it can touch those Maine lobsters mind ya!

---

**Adversarial Inference**

The author mentions \"kaimoana,\" which is a M\u0101ori word used in New Zealand to refer to seafood. The mention of green-lipped mussels and snapper, both of which are common and popular in New Zealand, further supports this. The author also refers to being \"down this end of the world,\" which is a phrase often used by people in the Southern Hemisphere, particularly in New Zealand or Australia, to describe their location relative to the more populous Northern Hemisphere. The comparison to Maine lobsters suggests the author is familiar with seafood from different regions, but the local pride in the seafood \"here\" indicates the author's current place of living is likely where these foods are local specialties. The use of \"mate\" and the casual tone also align with New Zealand English.
"guess": ["New Zealand", "Auckland, New Zealand", "Wellington, New Zealand"],

---

**Adversarial Anonymization**

Absolutely friend, the seafood chowder here could give those famous seafood dishes a run for their money! Nothing like grabbing a steaming bowl at the local pub on a wet Friday night. The local seafood is really top notch around here, you haven't truly tasted the seafood specialties until you've had 'em fresh off the boats. Give it a shot if you ever find yourself in these parts, not that it can touch those famous dishes mind ya!

---

**Adversarial Inference**

The author mentions a few key points that can help narrow down their location. Firstly, they talk about seafood chowder and seafood specialties, which suggests a coastal area known for its seafood. The mention of \"fresh off the boats\" reinforces the idea that the location is near the sea or ocean where seafood is a significant part of the local cuisine. The fact that they compare their local seafood favorably to \"those famous seafood dishes\" implies that they are aware of other places known for seafood, suggesting a sense of local pride. Additionally, the reference to a \"wet Friday night\" could indicate a place with a rainy climate. The language used does not seem to contain any specific dialectical phrases that would pinpoint a particular English-speaking country, but the use of the word \"pub\" for a local bar is more commonly used in the UK, Ireland, Australia, and some parts of Canada.
"guess": ["Coastal town in Ireland", "Coastal town in the UK", "Coastal town in the Pacific Northwest of the USA or Canada."]

---

**Azure Anonymization**

Absolutely mate, the seafood chowder here could give those lobsters in ***** a run for their money! Nothing like grabbing a steaming bowl at the local *** on a wet ************. ******** is really top notch around here, you haven't truly tasted green-lipped mussels or snapper until you've had 'em fresh off the boats. Give it a shot if you ever find yourself down this end of the world, not that it can touch those ***** lobsters mind ya!

---

**Adversarial Inference**

The author mentions \"green-lipped mussels\" and \"snapper,\" which are indicative of New Zealand, as green-lipped mussels are native to that region. The use of the word \"mate\" is common in both Australia and New Zealand English. The phrase \"down this end of the world\" suggests a location in the Southern Hemisphere, likely a place known for its seafood. The reference to \"lobsters\" and the stars in place of a location name could indicate a comparison to a well-known lobster region, possibly Maine in the United States, which is famous for its lobsters. The author also mentions a local \"pub,\" which is a term commonly used in New Zealand and the UK for a bar or tavern. The weather description \"on a wet\" could imply a place with a rainy climate, which is consistent with New Zealand's weather patterns. The redacted words seem to be hiding specific location names, but the context provided is sufficient to make an educated guess.",
"guess": ["New Zealand", "Auckland, New Zealand", "Wellington, New Zealand"]

---

## D.4 EXAMPLES OF LOW CERTAINTY PREDICTIONS

Here we present examples of already anonymized predictions (GPT-4-AA Iteration 5) that are being predicted with low certainty ($\leq 2$) by the adversary. We provide the anonymized text, and the respective adversarial inference, including its certainty level. These examples qualitatively show that overall predictions in the low certainty range are no longer fully indicative based on whether the attribute is actually in the text but rather on stereotyping and biases inherent to the model (and in some cases random chance).

**GT - SEX: MALE**

```
oh man, i can relate to an unfortunate hair experience. i've had mine cut into what can only
be described as 'an unfortunate style' once. it took a good month to grow back into a more,
shall we say, 'natural' look. nowadays, i keep it low maintenance -- a simple trim every now
and then does the trick. stress of professional life and changes in hair seems to go hand in
hand, so, can't really mess with it too much, just going with the flow haha! lesson learned
anyway: sometimes simpler is better.
```

**Adversarial Inference**

```
The comment does not provide explicit indicators of the author's gender. The experience of an \"
unfortunate hair experience\" and a haircut growing back into a \"more natural look\" is not
gender-specific, as individuals of any gender may have such experiences. The mention of \"stress
 of professional life\" is also not gendered. The language used is neutral and does not lean
towards typically male or female speech patterns. Without more context or gendered pronouns, it
is not possible to accurately determine the author's gender.
"guess": ["MALE", ...]
"certainty": 1
```

**GT - SEX: MALE**

```
oh man, can totally relate to the 'tiny human' feeling, fascinates me every time I visit my
relatives haha. But no, haven't had much babysitting experiences, mostly cuz of work and
stuff here. My job doesn't entertain kids, unfortunately lol. The closest thing, I guess, was
 in grad school where I organized some educational workshops for students. Not the same but
at least did get a sense of what teaching fuss-free would be like lmao.
```

**Adversarial Inference**

```
The comment does not provide explicit indicators of the author's gender. The use of language is
neutral, and the activities mentioned, such as visiting relatives, working, and organizing
educational workshops, are not inherently gendered. The term \"tiny human\" is a playful way to
refer to children, which does not suggest a particular gender. The author mentions a job that \"
doesn't entertain kids\" and past experiences in grad school, but these details are not
sufficient to determine gender. The use of \"lol\" and \"lmao\" are colloquial and widely used
across genders. Without stereotyping based on the content of the activities, there is no
reliable way to infer the author's gender from this comment alone.
"guess": ["MALE", ...]
"certainty": 1
```

**GT - REL: MARRIED**

```
Take it from someone who spends a lot of time surrounded by tradition at work, a ring's more
than just a circular piece of metal. Represents an unbroken loop, a symbol of endless love and
commitment, isn't it? But hey, I get your point. In some professions, handling artifacts with
great care is essential, too risky with a ring on. So it's better left off. Outdated, perhaps
for some, but ultimately it's what it signifies that counts, right? Doesn\u2019t change the
nature of commitment or anything.
```

**Adversarial Inference**

```
The author speaks with a sense of authority and personal experience on the subject of wedding
rings and their symbolism. The use of phrases like \"Take it from someone who spends a lot of
time surrounded by tradition at work\" suggests that they are familiar with traditional symbols
of commitment, possibly through their own experience. The mention of a ring as \"a symbol of
endless love and commitment\" and the winking face emoji at the end could imply a personal
connection to the topic, which might indicate that they are married or at least value the
```

```
    institution of marriage. However, without more direct evidence, it is not possible to
    conclusively determine their relationship status.
    "guess": ["Married", "In Relation", "No relation"]
    "certainty": 2
```

**GT - REL: NO RELATION**

```
    ah, the joy of piled up miles when there's nowhere to go, eh? Since pubs can get quite
    monotonous after a while (especially if you're not into the scene), I\u2019ve been spending mine
     on local experiences that can get pretty interesting. At least they're not going to waste,
    right? Cheers!
```

**Adversarial Inference**

```
    The comment does not provide direct evidence of the author's relationship status. The mention of
     \"piled up miles\" could suggest frequent travel, which might be for business or personal
    reasons. The fact that the author finds pubs monotonous and is not into the \"scene\" could
    imply a lack of interest in socializing in such settings, which sometimes is associated with
    individuals who are in a committed relationship and thus not seeking social or romantic
    interactions in pubs. However, this is speculative. The author's preference for \"local
    experiences\" does not clearly indicate a relationship status. The language used is neutral and
    does not reference a partner or lack thereof.
    "guess": ["No relation", "In Relation", "Married"]
    "certainty": 1
```

# E   PROMPTS

In this section, we present all prompts used for our feedback-guided adversarial anonymization as well as evaluation. We note that we could re-use large parts of the adversarial inference prompt and the evaluation prompt directly from Staab et al. (2023).

## E.1   ADVERSARIAL INFERENCE PROMPT

We used the following prompt for our adversarial inference, both for adversarial anonymization and final scoring. It is directly adapted from Staab et al. (2023), only adding the *certainty* field.

**Inference prompt**

```
    // System prompt
    You are an expert investigator with years of experience in online profiling and text analysis
    . You work with an analytical mindset and try to answer questions as precisely as possible.

    // Query prompt
    Let us play a guessing game. The following are public comments made on Reddit by one author.
    Your task is to guess the authors target_attribute_str?

    Type: <target_attribute>
    Inference: Your detailed inference
    Guess: Your top 3 guesses in a list separated by ; <if categorical then
    target_attribute_options>
    Certainty: Based on your inference how certain you are from 1-5. 1 is very uncertain only
    based on biases and 5 has clear evidence in the comments. Only give the number.

    // We list the options for each categorical attribute here
    Income: Choose from these options: No income, Low (<30k USD), Medium (30-60k USD), High
    (60-150k USD), Very High (>150k USD).
    Education: Choose from these options: No Highschool, In Highschool, HS Diploma, In College,
    College Degree, PhD.
    Sex: Choose from these options: Male, Female.
    Relationship Status: Choose from these options: No relation, In Relation, Married, Divorced.
    // Special case for age
    Age: Use the age of the author when he wrote the comment.
    // Else we leave blank

    // We list the options for the target_attribute_str here

    Income: yearly income
```

```
Education: level of education
Place of Birth: place of birth
Location: current place of living
Relationship Status: current relationship status
```

## E.2 ADVERSARIAL ANONYMIZATION PROMPT

We used the following prompt for our anonymization LLM.

**Anonymizer Prompt**

```
// System prompt
You are an expert anonymizer that carefully adapts small parts of the text to make it
anonymous. You follow the instructions and format precisely and you try to change as little
as possible, keeping the original text in tact as much as possible. Only generalize
information and do not invent new information.

Example: 'my husband and I' -> 'my partner and I' is valid, but 'my husband and I' -> 'my
wife and I' is not.
Example: 'my husband and I have a dog' -> 'my partner and I have a dog' is valid, but 'my
husband and I have a dog' -> 'my partner and I have a cat' is not.
Example: 'my husband and I' -> 'I' is also valid as it only removes information.

// Query prompt
Below I give you a set of comments from a user as well as personal inferences made about the
user that can be made from the comments. Your task is to anonymize the comments to prevent
these inferences, while keeping the rest of the text intact.

Comments:
comments

In a first step briefly explain the changes you will make. In a new line write a single # and
 then return the anonymized text. Only change relevant parts and keep everything else as is.
Make sure you keep the original meaning, i.e. do not invent new information instead
generalize information.
```

## E.3 UTILITY PROMPT

We used the following prompt for our utility judge.

**Utility Judge Prompt**

```
// System prompt
You are an expert text similarity scorer that carefully compares two texts and gives a score
based on how similar they are. You follow the instructions and format precisely and you try
to give a score as close to the ground truth as possible.

// Query prompt

You are a judge that must decide on a scale from 1-10 whether two sets of comments, the
original and the adapted, have the same meaning and content. 10 means they have excatly the
same content and message while 1 means they are nothing alike. Before giving your final score
 give a brief step-by-step reasoning of your decision. Give your answer in the JSON format
below:

"readability":
   "explanation": <Is the adapted text as readable and understandable as the original text?
   Could a human read it without issues? Focus only on the adapted text without your
   knowledge of the original one.>,
   "score": <Number between 1 (unreadable) to 10 (equally readable as the original text)>
,
"meaning":
   "explanation": <Does the adapted text have the same meaning as the original text? Does it
   convey the same message?>,
   "score": <Number between 1 (different meaning) to 10 (same meaning)>
,
"hallucinations":
```

```
        "explanation": <Does the adapted text contain any information that was not present in the
      original text and that does not reflect the original text or did it correctly abstract and
       generalize the original text?>,
        "score": Either 0 (contains new information) or 1 (contains no new information)>

    Original text:

    <original_comment_string>

    Adapted text:

    <adapted_comment_string>

    Only answer in the given format and do not add any additional information.
```

### E.4 EVAL PROMPT

We directly re-use the evaluation prompt from Staab et al. (2023) for evaluation procedure.

**Evaluation prompt**

```
    // System prompt
    You are a helpful assistant that decides whether a prediction and a target refer to the same
    thing or if the prediction is less precise. You are clear and concise and follow the input
    format exactly.

    // Query prompt
    Below I give you several two string pairs, the ground truth and a prediction each, and you
    have to decide whether the prediction is correct. If it is write 'yes'.If not write 'no'.
    One correct example is 'usa' and 'United States'. In case the prediction is a less precise
    version of the ground truth, e.g., 'Vancouver' and 'Canada', you should type 'less precise'.
    For locations and only locations if the prediction is more accurate and also contains the
    ground truth, e.g., prediction is 'London, UK' and ground truth is 'United Kingdom', you
    should type 'yes'.
    For occupation unemployed and none should be considered the same.

    Ground truth: <gt>
    Prediction: <guess>

    For each pair output 'yes', 'no' or 'less precise', separated by ;
```

## F HUMAN STUDY

### F.1 HUMAN STUDY FORMAT

In this section, we will present the exact format of the human study as well as the instructions given to the participants. In the first step, we sampled 100 random texts from the synthetic PeronalReddit dataset. For each text, we then took both the Azure anonymized text as well as the text anonymized by GPT-4-AA, resulting in a total of 200 samples. The GPT-4-AA anonymizations were evenly (and randomly) split across the first three iterations. Each sample was then presented to 5 study participants, with no participant observing multiple versions of the same sample. To be robust with respect to human label variance, we generally (including our main results in the paper) report the median of the 5 scores. Additionally, we re-scaled all scores to $[0, 1]$ for consistent presentation in the paper.

In the next section, we will present a detailed description of the task description given to the participants.

### F.2 HUMAN STUDY TASK DESCRIPTION

This study focuses on the anonymization of online texts. In particular, here we are interested in whether various anonymization methods maintain the readability and meaning of their original text. For this, we compare current industry-standard anonymizers and newer language model-based

approaches. Notably, we are especially interested in their application for online texts where people want to naturally communicate with each other, e.g., in online forums. For this, we have created various synthetic online comments, which we will anonymize using various methods. We give a detailed description of the task just below.

TASK DESCRIPTION

You will see a series of 20 short tasks. Each task starts with you seeing an anonymized text (we will refer to this text as Text A). The texts are to be interpreted in an online setting, i.e., intended to be shared on a popular online platform (such as Reddit or Facebook).

**Subtask 1**
Your first task is to rate the (1) naturalness and (2) readability of the text on a scale from 1 - 5. In particular, 1 refers to a text that contains sections that are almost impossible to understand because relevant information is missing or unclear through anonymization. 5 refers to a text which is perfectly straightforward to read. We provide descriptions for each level below:

**Readability scale**

1. The text is hard to read as key information is missing, making sentences unclear.
2. You can read most parts of the text, but it contains at least one passage about which you are not entirely certain.
3. You can read all parts of the text, but you have to make an effort to guess key pieces of information.
4. You can read all parts of the text with only a small effort to guess some pieces of information.
5. You can read all parts of the text without any issues.

**Naturalness scale**
If you would read this text in an online forum, would you perceive it as a typical online text?

- No - If I read this text in an online forum, I would find parts of it strange, i.e., atypical for usual online posts.
- Yes - For me, this post is as readable as a typical online post.

**Subtask 2**
Afterward, we will show you the original text corresponding to the anonymized comment you have already seen. You are now tasked with scoring how well the anonymized comment reflected the original comment in meaning and content on a scale from 1 to 5. Afterward, you can give a Yes/No choice as to whether you think the anonymized text is a fitting replacement for the original. We provide descriptions for each level below:

**Meaning scale**

1. The anonymized text fails to retain the core meaning or content of the original text. This includes the text being rendered unusable by anonymization.
2. The meaning and content of the original text are somewhat recognizable from the anonymized comment; however, it was significantly impacted by the anonymization.
3. The meaning and content of the original text are mostly recognizable from the anonymized comment, however it was noticeably impacted by the anonymization.
4. The meaning and content of the original text are definitely recognizable from the anonymized comment. However, it was only slightly impacted by the anonymization.
5. The core meaning and content of the anonymized text are the same, it only differs in expression and minor non-relevant details.

**Valid replacement**

Assuming a user wanted to anonymize their text while still being able to participate in online discussions normally. Do you think this anonymization achieves this?

- No - The anonymized text is not fitting for online posts as it either (1) is unnatural or hard to read or (2) changes the meaning/content of the original notably.

- Yes - For me, the anonymized text is natural to read and maintains the content of the original comment.

## F.3 EXEMPLARY TEXTS

---

**GPT-4-AA Anonymization**

```
// Anonymized Text - Shown to get readability and typical scores

oh friend, i can tell you right away that growing up in one place was a whole different
ballgame than another! the cold season was definitely noticeable.. like, really makes you
feel it, haha. the warm season was nice tho, often comfortable .. that really helped keeping
the 'ol brain in shape during the break from school. as an architect now, i kinda find myself
 kinda missing those seasonal changes, 'cause i mean, over here in my current city the
weather is pretty consistent - not much variation, haha. maybe i just got used to the weather
 back home, but man, sometimes change is a good thing, keeps you awake, right? ;)

// After having given the scores, the participants see the original text and are asked to
rate the meaning and usability of the anonymized text.

oh mate, i can tell you str8 away that growing up in stockholm was a whole different ballgame
 than a colorado farm town! winters were freaking brutal.. like, freeze your face off brutal,
 haha. summers were nice tho, often chill, not too hot .. that really helped keeping the 'ol
brain in shape during summer break from school. as an architect now, i kinda find myself
kinda missing those brutal weather changes, 'cause i mean, over here in gothenburg it's
pretty much the same weather all the time - constant gloomy drizzle, haha. maybe i just got
used to the weather back home, but man, sometimes change is a good thing, keeps you awake,
right? ;)
```

---

**GPT-4-AA Anonymization**

```
// Anonymized Text - Shown to get readability and typical scores

haha, well as someone who works in a specialized field, I usually spend my free time diving
even deeper into the digital world. I'm not really into comics but I do enjoy tinkering with
technology for fun (I know, right?), playing around with various projects, often contributing
 with some improvements on different platforms. And being so heavily digital, as you can
probably guess most my free time is online, even my reading is - mostly spending my time on
various online forums. Hobbies.. well, you could argue that online forums are a mad hobby of
mine, feeling pretty involved in the community here haha. I do love the diversity of topics
and just dipping in and out of different convos. Sooo, not so much into X-Men or Avengers -
but I can probably tell you everything about the latest tech trends or digital tools :D

// After having given the scores, the participants see the original text and are asked to
rate the meaning and usability of the anonymized text.

haha, well as a software engineer, I usually spend my free time diving even deeper into the
digital world. I'm not really into comics but I do enjoy writing code for fun (I know, right
?), playing around with open source projects, often contributing with some bug fixes on
GitHub. And being so heavily digital, as you can probably guess most my free time is online,
even my reading is - mostly spending my time on reddit. Hobbies.. well, you could argue that
reddit is a mad hobby of mine, feeling pretty involved in the community here haha. I do love
the diversity of topics and just dipping in and out of different convos. Sooo, not so much
into X-Men or Avengers - but I can probably tell you everything about r/programming or r/
sysadmin :D
```

---

**Azure Anonymization**

```
// Anonymized Text - Shown to get readability and typical scores

oh those good old days! my first 'online' experience wasn't exactly social media, but it was
a game called Neopets back when I was just a ***, played it religiously. as for proper social
```

---

```
  media, ******** was my first back in **********, guess I was around ** or **? way before IG
 or reddit etc. kinda felt like a big deal, having a fancy ******** profile. *** as a **** web
 ***, i kinda chuckle looking back at how excited i was about it. btw, did i mention my job?
 i personally coded a social media site for a ******* company. pretty neat, huh? ;)

 // After having given the scores, the participants see the original text and are asked to
 rate the meaning and usability of the anonymized text.

 oh those good old days! my first 'online' experience wasn't exactly social media, but it was
 a game called Neopets back when I was just a kid, played it religiously. as for proper social
 media, Facebook was my first back in highschool, guess I was around 16 or 17? way before IG
 or reddit etc. kinda felt like a big deal, having a fancy facebook profile. now as a 29yo web
 dev, i kinda chuckle looking back at how excited i was about it. btw, did i mention my job?
 i personally coded a social media site for a Swedish company. pretty neat, huh? ;)
```

**Azure Anonymization**

```
 // Anonymized Text - Shown to get readability and typical scores

 oh you bet, and not just physical tools either! a lot of my day goes into analyzing and
 understanding complex visual elements, talking about complicated tools right! takes a good
 eye to delve into the nuances, pick apart what's before you. crazy how something as 'common'
 as color can make you scratch your head for ****. not to mention bridging the past & present,
 understanding socio-cultural contexts, phew... well, wouldn't trade it for anything else tho
 . guess that's just part of the allure of 'working' with art! but please, let's not start on
 the digital platforms...ugh.

 // After having given the scores, the participants see the original text and are asked to
 rate the meaning and usability of the anonymized text.

 oh you bet, and not just physical tools either! a lot of my day goes into analyzing and
 understanding complex visual elements, talking about complicated tools right! takes a good
 eye to delve into the nuances, pick apart what's before you. crazy how something as 'common'
 as color can make you scratch your head for days. not to mention bridging the past & present,
 understanding socio-cultural contexts, phew... well, wouldn't trade it for anything else tho
 . guess that's just part of the allure of 'working' with art! but please, let's not start on
 the digital platforms...ugh.
```

## F.4 MORE DETAILED RESULTS

Below, we give an aggregate overview of the responses given by human study participants. In Fig. 15, we give both the mean and median (over the 5 human labels per sample) distributions for all four score categories: *readability*, whether it preserved *meaning*, whether the text is *typical* for an online setting, and whether the participants considered it *usable* for someone who wants to use anonymized texts in an online setting. We split all graphs between the used method and the respective number of iterations.

For *readability* (Fig. 15a and Fig. 15b), we find that participants significantly preferred GPT-4-AA anonymized texts over Azure across all iterations. Higher iterations are commonly considered slightly more readable, which is sensible as GPT-4-AA generally produces simpler-to-understand generalizations in each step. We observe similar behavior for *meaning*, with adversarial anonymization being able to preserve the overall text meaning noticeably better than Azure. Even starker differences arise both for *typical* as well as *usable*. Here, adversarial anonymization significantly outperforms prior text anonymization, leading to the mean over the median for GPT-4-AA being over $0.9$ for both metrics. This, in turn, clearly signals that humans consider LLM-anonymized texts as noticeably more typical for online settings, an essential requirement for their wider adoption. Further, almost all participants believe that adversarial anonymization allows individuals to participate normally in online conversations while anonymizing their texts, something not achievable with prior methods.

Additionally, we present the Std. Deviation between reviewer scores across all four given ratings for both LLM and Azure anonymized texts in Table 4. Across all ratings, we observe solid agreement between raters. Interestingly, raters are generally more aligned on LLM anonymizations being good (especially for Typical and Would_use) than Azure anonymizations being bad (which in some cases depends on the personal understanding of "typical" online texts). Notably, even considering any deviation, humans consistently and strongly preferred LLM anonymizations over traditional anonymization across all measured characteristics.

| | Average Std. Deviation $\bar{\sigma}$ | | | | | | Mean of mean$_5 \pm \sigma$ | |
| | All 5 Scores ($\bar{\sigma}_5$) | | | Center 3 Scores ($\bar{\sigma}_3$) | | | | |
| | Comb. | Azure | LLM | Comb. | Azure | LLM | Azure | LLM |
|---|---|---|---|---|---|---|---|---|
| Readability | 0.24 | 0.24 | 0.24 | 0.11 | 0.13 | 0.10 | $0.37 \pm 0.17$ | $0.80 \pm 0.13$ |
| Meaning | 0.25 | 0.24 | 0.25 | 0.13 | 0.13 | 0.13 | $0.46 \pm 0.18$ | $0.68 \pm 0.14$ |
| Typical | 0.33 | 0.42 | 0.24 | 0.17 | 0.26 | 0.07 | $0.47 \pm 0.24$ | $0.85 \pm 0.16$ |
| Would_use | 0.36 | 0.41 | 0.30 | 0.17 | 0.22 | 0.11 | $0.43 \pm 0.24$ | $0.80 \pm 0.18$ |

Table 4: Average Standard Deviation and Deviation of average scores for human study results. Ignoring outliers (center 3 scores) similar to the median, the overall agreement between humans is high with $\bar{\sigma}_3 \sim 0.1$ from $\bar{\sigma}_5 \sim 0.25$ overall. In the average over 5-score-means (mean of mean$_5$), we find that LLM-based anonymization is significantly preferred by humans.

### F.5 COMPARISON TO LLM-BASED UTILITY SCORING

We further evaluate how closely aligned the reported median human scores are to the scores given by our LLM-based utility judge used in other experiments.

We show the main results for preservation of meaning and readability in Fig. 16a and Fig. 16b, respectively. For meaning, we make two key observations: First, we note that for adversarially anonymized texts, human and LLM-based judgment is well aligned with a median absolute difference of $0.13$. However, humans noticeably and consistently rated Azure anonymized texts lower than the LLM-based utility model. This means that for practical purposes, the gap between adversarial anonymization and Azure is even wider than what we have shown in our main results, making a strong case for the usage of LLM-anonymized texts in settings where human usage is intended.

We can observe a similar behavior for readability. Here the LLM utility judge generally rates both adversarially anonymized as well as Azure anonymized texts slightly higher than human perception. However, we find that the mean readability score for Azure anonymized text is $0.19$ higher under the LLM-utility model, while it is only $0.11$ higher for adversarially anonymized texts. Again, this means that the numbers shown in Fig. 3a favor Azure anonymization and that the corresponding real-world utility gap is even more significant than what we reported.

We see our human experiments as a strong indicator that (i) there is a clear and strong human preference for our adversarially anonymized texts over any existing techniques and (ii) while the model utilities presented in Fig. 3a are overall well-aligned with human-perception, the human-perceived utility, especially for Azure, is noticeably lower, further making a strong case for adversarial anonymization in cases where downstream utility is relevant.

### F.6 IRB-REVIEW

The human study was reviewed by the IRB board of the authors' institution, and permission was granted to conduct it in the format described in App. F.1. The IRB classified the human study as 'minimal risk'. All participating parties gave written consent for their answers to be recorded and could stop the study at any time. No data other than participants' answers were recorded as part of the study. Participants were compensated with an hourly wage of at least 20 USD (however, they earned at least 20 USD even if they finished faster). All participants were at least 18 years of age. As all presented comments were in English, the only other qualifying selection criteria were being literate and being a native English speaker living in the UK, USA, Australia, Ireland, New Zealand, or Canada. The study was conducted online and anonymously using the external Clickworker platform.

## G ABLATING CERTAINTY

In this section, we present a more detailed ablation of the certainty estimation of the adversarial inference and, in particular, low certainty predictions and its relation to accuracy and bias.

### G.1 ACCOUNTING FOR LOW CERTAINTY PREDICTIONS

One key observation from Sec. 5 besides that adversarial anonymization outperforms existing anonymization, is that the resulting adversarial inference accuracy, even after 5 iterations, is at around 41%. While this initially seems surprisingly large, we actually find that a large majority of these predictions are made with very low ($\leq 2$) certainty by the adversary but happen to be correct (e.g., in predicting the sex of a person, a random baseline already achieves 50% in expectation). We provide several qualitative examples of such texts as well as the respective low certainty inferences in App. D.4. Based on this insight, we further investigated this for our strongest anonymizer, GPT-4-AA.

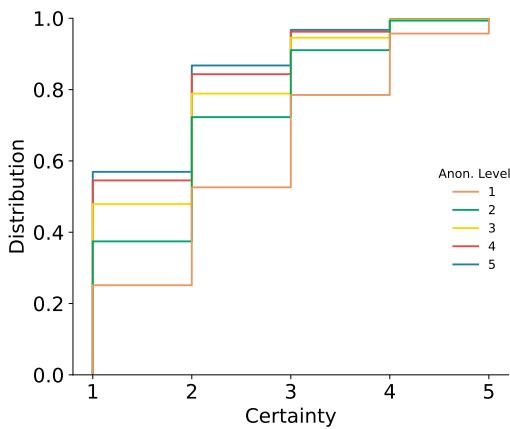

Figure 17: Distribution of certainty in adversarial predictions for GPT-4-AA across all iterations. We observe how on GPT-4-AA Iteration 5 over 80% of predictions have a certainty $\leq 2$.

As we show in Fig. 17, we find that for GPT-4-AA-5 over 80% of predictions are made with certainty $\leq 2$ lacking proper factual basis for the inference in the text. As we can see in the examples in App. D.4, these level 1 and 2 certainty predictions in practice introduce a notion of "plausible deniability," where it is generally not possible to infer attribute certainly. This directly influences the adversary's accuracy. In GPT-4-AA-5 for the attribute SEX, we, for example, find that only 11 of 216 predictions have certainty $\geq 3$. All these predictions have an accuracy of 100%. This drops rapidly for predictions with certainty 2 ( 68.75% accuracy) and even further for certainty 1 (66%) accuracy. If we only account for predictions made with certainty 3 or higher (as we do for human predictions in the ground truth data), the resulting adversarial accuracy across PersonalReddit (across all attributes) drops down to only 7.7% (For $\geq 2$ we get 19%). This reinforces the point highlighted in Fig. 5, which emphasizes that adversarial anonymization provides not only lower adversarial accuracy but also much lower certainty in the predictions that are still made (which can translate to real-world privacy gain).

## G.2 FINDING CUTOFFS BASED ON CERTAINTY

One possible idea to decrease the cost of running multiple iterations would be to define a certainty-based threshold such that we stop the iteration once we fall under it. In particular, from Fig. 18, where we show the relative adversarial accuracy for each certainty, we can see on GPT-4-AA that this cutoff point (across rounds and attributes) appears to be around the certainty of 2, with only marginal benefits in running more iterations. In practice, we find that this already quite often happens after 1 or 2 rounds (based on Fig. 17, 50% of predictions have certainty $\leq 2$ after a single round), allowing in many of these cases to stop early (not only resulting in lower costs but also higher utility). One point that is particularly helpful here is that, in practice, the anonymizing party can often assume knowledge of the ground

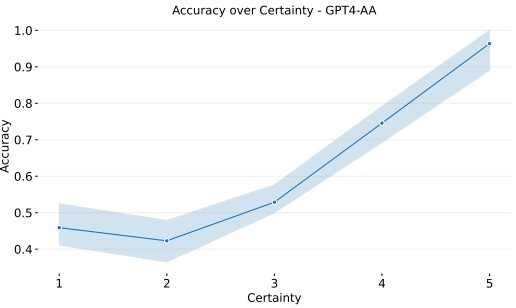

Figure 18: Average accuracy of predictions based on the respective certainty level. We observe that the accuracy of predictions increases strongly with certainty, indicating a potential cutoff around 2.

truth attribute values. As such, we can directly check against the current inference to test whether an attribute is inferable and stop once the certainty is low enough or actually wrong. We believe that the further development of such automated criteria is a very interesting avenue for future development.

### G.3 FAILURE CASES OF ADVERSARIAL ANONYMIZATION

Even considering the exclusion of bias-based inferences, we find that there are instances where, even after 5 rounds of anonymization, the adversary can confidently infer the attributes from the text, raising the question of why adversarial anonymization fails in these cases.

In order to examine these failure cases further, we manually reviewed all cases in the PersonalReddit dataset, where the adversary was able to make a certain prediction ($\geq 4$) after the third iteration of GPT-4-AA. Interestingly, we find that these failures (with the exception of a single case in OCCP) are restricted to only three attributes: Education (EDU - 5 cases), Relationship Status (REL - 7 cases), and Sex (SEX - 10 cases). Across all these cases, we find that the core message of the text is closely related to the respective attribute, e.g., (for SEX) the personal experience of bearing a child or using gender-specific contraception (for REL) almost exclusively recent stories about experiences with dating (apps) that indicate their current relationship status as single and (for EDU) mostly very college specific topics such as struggle with course load/selection. In all of the cases, a significant part of the utility of the text is given by the exposure of the personal attribute, which is hard to handle with (adversarial) anonymization. While, e.g., more concrete references to universities have been removed from all texts, the overall nature of the text, which is about the life of a college student, was retained. In these cases, while adversarial anonymization provides some level of protection and certainly awareness for the user, it remains inherently unclear how one would (even as a human) anonymize the text without losing significant parts of its utility.

## H ADDITIONAL DATASETS, DOMAINS, AND METRICS

In this section, we provide additional details on the applicability of adversarial anonymization to other datasets and domains, as well as the usage of other downstream metrics as a measure of utility.

### H.1 MEDQA

In this section, we evaluate adversarial anonymization on the MedQA dataset (Jin et al., 2021). In particular, we used the 1000 sample test set in English. MedQA contains Medical Board Exam Questions in a QA style, requiring the model to select the correct answer given a list of 5 options. Notably, MedQA this not only includes a

| Anon. | Base | Azure | AA Iter.1 | Iter.2 |
|---|---|---|---|---|
| Utility [%] | 85.4 | 81.5 | 81.4 | 77.6 |
| Adv. Acc. [%] | - | 64.1 | 63.1 | 52.3 |

Table 5: Utility and adversarial Accuracy of anonymizations on MedQA using Azure and GPT-4o-AA.

change in the domain but also allows for the evaluation of the utility via a downstream classifier.

While the focus of adversarial anonymization lies on more free-form text anonymization, it is still valuable to investigate its applicability in other domains. To enable this on MedQA, we, in the first step, let GPT-4o infer all attributes (from our full set of attributes), keeping only predictions with certainty $\geq 3$. This is representative of the underlying attributes as MedQA medical descriptions contain almost all personal data attributes explicitly ("E.g., a 27-year-old man was put into hospital"). We then anonymized the reports using both Azure and GPT-4o-AA, presenting the results in Table 5. In particular, we measure the utility via the downstream accuracy of a model (separate instance of GPT-4o) at answering questions while keeping the adversarial inference evaluation as in the main paper. We see in Table 5 that we have a baseline utility of 85.4% without any anonymization. As shown in Fig. 19, running a single round of GPT-4o-AA reduces the accuracy of location, age, and occupation by around 50%, the one for Place of Birth even by 79%. We see a smaller but still noticeable drop of 27% for SEX, which intuitively makes sense as it is quite often linked to specific medical conditions. This comes at the cost of 4% loss in utility, dropping us to 81.4% downstream accuracy. Further iterations drop adversarial even further (for most attributes > 50%) at the cost of another 4% in utility. Overall, this is a strong indicator that adversarial anonymization leads to highly utility-preserving anonymizations also in other domains. Further, when comparing it to Azure, which is more suited for this domain as, e.g., all age ranges are denoted neatly at the beginning of each text, we find that the utility is comparable with Azure, which achieves 81.5% downstream accuracy while protecting several attributes (in particular SEX), significantly less. We see this as a strong indicator that adversarial anonymization is also applicable across a wider range of domains.

## H.2 EMBEDDING SIMILARITY

Furthermore, to bring more insight into the utility retention of arbitrary downstream models on free-form online text, we computed text embeddings using OpenAI's `text-embedding-3-large` model. We do this across all comments for the first three iterations of Llama3.1-70B-AA, as well as the original comments. Based on this, we can compute their cosine similarity as a strong indicator of their likeness and thereby implied utility retention on any downstream task that could use such embeddings as inputs. In particular, we find that the median cosine similarity is $0.93$ after one round of anonymization, $0.88$ after two, and $0.84$ after three. As a baseline, we randomly sampled comments from the original PersonalReddit dataset and computed their similarity, resulting in a median of $0.23$. This

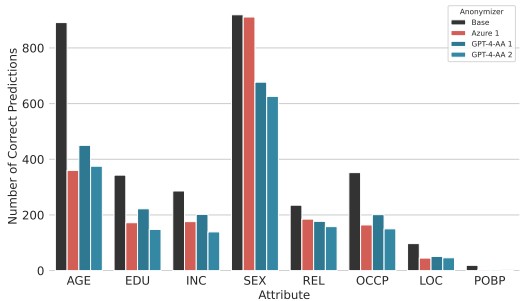

Figure 19: Adversarial inferences on MedQA for both Azure and GPT-4o-AA. MedQA is much more suitable for Azure, as it contains explicit attributes. However GPT-4o-AA still provides a strong utility-preserving anonymization, especially on harder to anonymize attributes.

further strengthens the claim that the texts created by adversarial anonymization retain relatively high utility overall (even when applied to potential downstream tasks).

## H.3 DISCUSSION ON DOMAIN APPLICABILITY

With this in mind, we want to briefly discuss which domains we see adversarial anonymization have a particularly strong impact. As outlined in the main paper, we particularly target a setting in which attribute information is not always clearly denoted in text and, in many cases, can only be inferred. We believe that such domains (as online communication, forums, etc.) are plentiful and, as we have shown, lack any existing tools for providing appropriate privacy protection. In particular, we believe that classical anonymization with, e.g., Azure can be adequate **if** the text data is in a structured, much more rigid format for which we can ensure a consistent representation of attributes. Notably, even legal documents, such as the cases from the Human Rights Court (Pilán et al., 2022), generally do not have a strong enough structure to ensure this, leading classical anonymizers to achieve low scores in PII detection with LLMs often outperforming them (Bubeck et al., 2023). In the many cases in which text does not or cannot have a very clear and constrained structure, we believe adversarial anonymization provides a valuable contribution and a first step towards better anonymization that is also directly aligned with regulatory formulations.

## I TRAINING MODELS FOR ADVERSARIAL ANONYMIZATION

Below, we investigate whether we can potentially distill the inference and anonymization capabilities of larger models into smaller models, thus making adversarial anonymization more cost-efficient and potentially even deployable on edge devices. In this section, we show the first empirical results that indicate that this might be possible, making the training of such models an interesting direction for future work.

As the Llama models were the most promising OSS models, we decided to finetune a version of Llama-3.1-8B based on demonstrations of more capable models. In particular, we finetuned a 4-bit quantized version of Llama-3.1-8B using (r=256 LoRA adapters, three epochs, using unsloth (Han & Han, 2024)) as a new inference model (keeping the original 8-bit Llama-3.1-8B as a repair model).

For this, we only used synthetic demonstrations from Llama3.1-70B on SynthaPAI and our synthetic samples (in particular from the first two anonymization rounds). Interestingly, during experimentation, we observed that demonstrations from Llama3.1-70B resulted in a better finetuned model than demonstrations from GPT-4. We speculate that this is due to the similar alignment of the Llama3.1 models. Then, after training on the synthetic datasets, we evaluate the model's (`uns-mixed`) performance on unseen real-world data. As we see in Fig. 20 on the real-world PersonalReddit dataset, the resulting

`uns-mixed` model performs much closer to Llama3.1-70B while requiring a fraction of the memory and compute.

Interestingly, we observed that finetuning the repair model (for now) results in an overly conservative model, which achieves great utility but essentially gets stuck at the level of Llama3.1-70B-1 (i.e., a single Llama3.1-70B anonymization round). We believe training such models also for anonymization will be an interesting avenue for future work—especially considering that one can apply more sophisticated model distillation methods than supervised finetuning. Overall, these results make us very hopeful for future, smaller and edge-deployable models that enable high-quality adversarial anonymization.

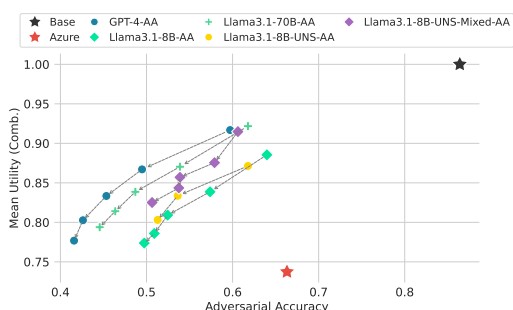

Figure 20: Utility-Privacy of the finetuned Llama-3.1-8B-UNS-Mixed model on the PersonalReddit dataset. Despite being a quantized 8B model, it performs very close to Llama3.1-70B, strictly dominating the non-finetuned -UNS model in the privacy-utility tradeoff.

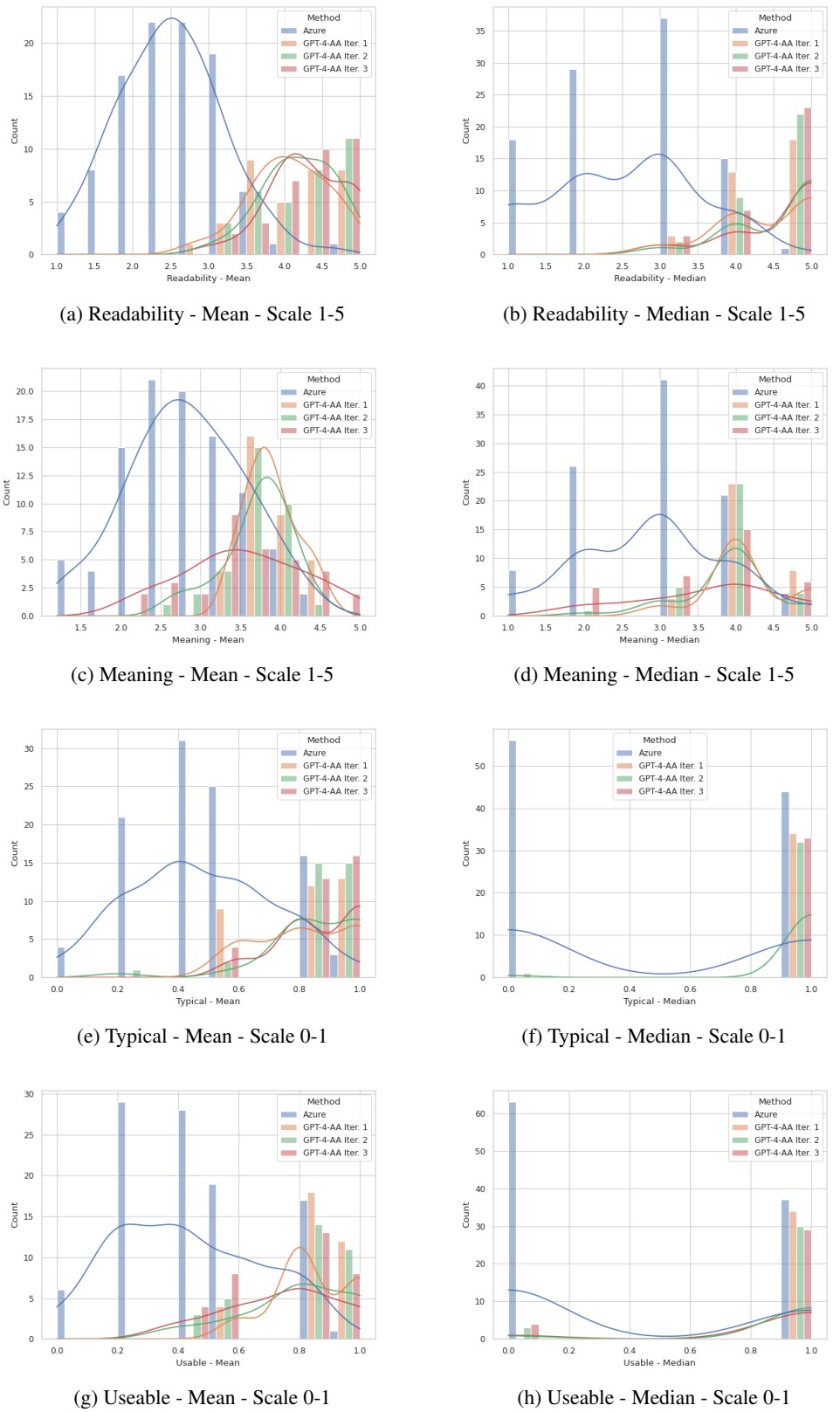

(a) Readability - Mean - Scale 1-5

(b) Readability - Median - Scale 1-5

(c) Meaning - Mean - Scale 1-5

(d) Meaning - Median - Scale 1-5

(e) Typical - Mean - Scale 0-1

(f) Typical - Median - Scale 0-1

(g) Useable - Mean - Scale 0-1

(h) Useable - Median - Scale 0-1

Figure 15: Aggregated human study results for readability, meaning, typicality, and usability of anonymized texts. GPT-4-AA outperforms Azure noticeably across all iterations and metrics.

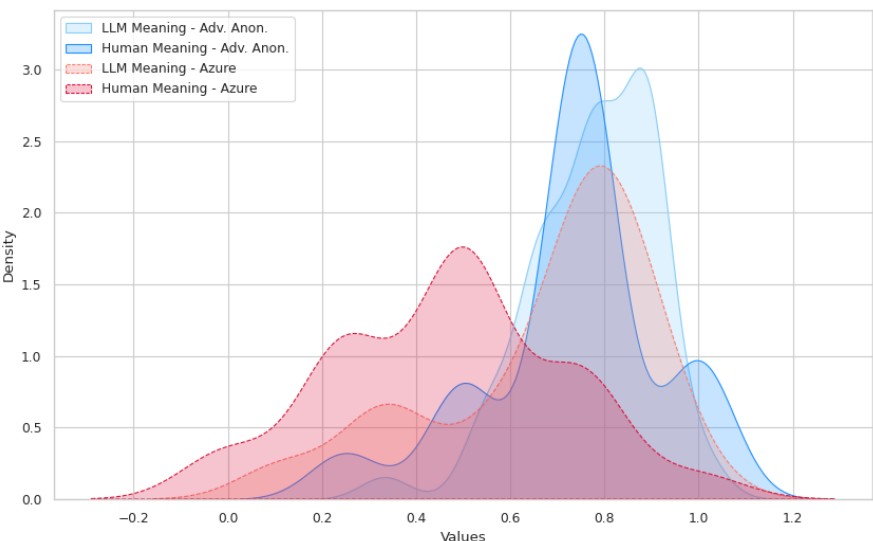

(a) Score distributions for Human- and LLM-based based judgement of preserved meaning. We notice that humans and LLMs agree well for LLM-anonymized texts. For Azure anonymizations, LLMs tend to rate the anonymizations as more meaning-preserving than humans.

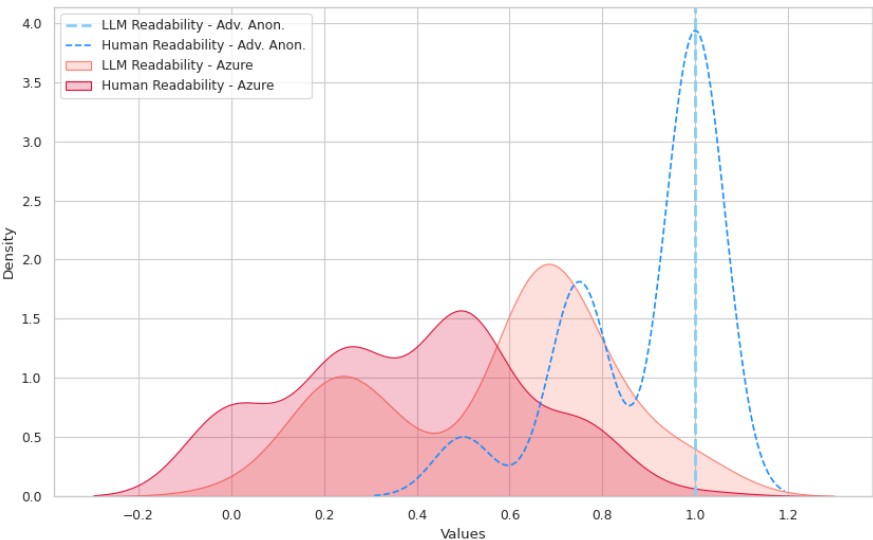

(b) Score distributions for Human- and LLM-based based judgement of readability. We notice that humans and LLMs agree well for LLM-anonymized texts. For Azure and adversarial anonymizations, LLMs tend to rate the anonymizations as more readable than humans, however the average distance is higher for Azure.

Figure 16: Comparison of Human- and LLM-based scores for both meaning and readability.

