# OpenReview forum: "Language Models are Advanced Anonymizers"
_ICLR.cc/2025/Conference — ICLR 2025 Poster_

### Official Review · Reviewer_18jd · 2024-10-27

**Soundness:** 3
**Presentation:** 3
**Contribution:** 2
**Rating:** 5
**Confidence:** 4

**Summary:**

This paper proposes a LLM-based adversarial anonymization framework to address privacy risks. The authors use a feedback-guided approach where an LLM adversary attempts to infer personal attributes from a given text, and an anonymizer LLM iteratively modifies the text to reduce inference risks. The paper evaluates this method against traditional anonymization techniques and demonstrates superior performance in both preserving utility and privacy across several datasets.

**Strengths:**

1. The proposed adversarial anonymization framework leverages the strengths of LLMs both as adversaries and anonymizers, showcasing a new application of LLMs in a privacy-preserving context.
2. The inclusion of a human study adds value to the evaluation by confirming the practical applicability of the framework and showing a preference for the LLM-anonymized text.

**Weaknesses:**

While this might be an interesting application of LLMs to the field of anonymization, the core methodology introduces neither fundamentally new anonymization techniques nor a different way to use LLMs. It merely adapts existing concepts by leveraging the powers of LLMs. Thus, the contribution of novelty is limited for either the LLM or the privacy community.

One major limitation is that, in real life, texts to anonymize are normally very long, and due to the current method, which would be prohibitively expensive, it is practically not feasible in applications that demand real-time processing. The scalability and applicability of this framework are rather limited due to the enormous amount of documents it may need to work with iteratively. Another limitation is that privacy performance remains unpredictable due to heavy dependence on the capabilities of the LLM. This creates a dependency where consistency of anonymization outcome cannot be guaranteed and may even differ from model to model or update to update.

**Questions:**

1. Could you provide more examples where the framework fails and explain why the LLM is unable to recognize these instances?
2. Do you think it’s feasible to distill this capability into a smaller model? in this way, we can reduce the computational cost.

---

> ### Author Response · Authors · 2024-11-22
>
> We thank the reviewer for their comprehensive feedback. We respond to the raised concerns below, numbered Q1-Q5. We have also uploaded a revised version of the paper, with updates highlighted in violet. We are happy to continue the discussion further if the reviewer has additional questions.
>
> **Q1: Does the simplicity of the final anonymization algorithm decrease the overall merit of the work?**
>
> No, we believe that even though our eventual anonymization method is conceptually straightforward from a purely technical perspective, it makes significant contributions in the field of text anonymization in several ways—especially so in casual communicative settings, such as online forums, where, as shown in [1] LLMs pose a severe novel threat to anonymity.
>
> On the setting side, we are the first to provide an (inference-based) adversarial view on text anonymization, addressing the key limitation of prior stance on anonymization; namely that even entity-blanking-based anonymization still leaks private information in presence of an inferential adversary. This is a critical issue that is fundamental to current anonymization tools and evaluation. As also shown in [1], we demonstrate in our evaluation that on real-world data existing methods are clearly insufficient both against adversaries and in utility.
>
> On a practical side, as reviewers agree, adversarial-feedback anonymization using LLMs provides a natural instantiation of this setting and achieves consistently better privacy protection and utility than *commercial* anonymization tools. As such, our strong empirical contributions could have an noticable impact on the state of anonymization currently in application.
>
> Further, having such an elaborate analysis and extensive experimental evaluation (across 17 models, multiple datasets, and including a human study), sets a strong baseline for any potential follow-up work in this area.
>
> As such, we believe that there is significant merit in (1) defining a practical and free form setting for text anonymization and (2) providing a method that outperforms current industry-standard anonymization across very extensive evaluations. Such a contribution is particularly timely as developments in LLMs have made such inferences much easier [1] up to the point where it can be potentially misused on a large scale [2], constituting a practical real-world threat.

---

> > ### Author Response · Authors · 2024-11-22
> >
> > **Q2: Why did you choose online communication as a setting? How does this deal with long context sizes and real-time processing?**
> >
> > We primarily focused on the online domain as (1) it is a setting where real-world risks were shown in prior works [1] and (2) it is a setting which is uniquely challenging for existing methods in anonymization—to an extent where they are borderline unusable (and unpreferred, as shown in our human evaluation). In that sense, we believe that providing a solution that works well here is a valuable contribution for potential users as well as the privacy community.
> >
> > In particular, the inferential privacy setting addresses core issues in prior work in text-based anonymization using span-based tags that required significant human effort to collect and combine [3], overall hindering process in this area [4].
> > Further, we do not believe that context size is a significant issue for (1) the primary use-case we are targeting (while online conversations are commonly shorter in length, PersonalReddit contains texts of up to ~1.5 pages (1000 tokens) - not just short individual comments) and (2) the significantly larger context size with many models already now being able to handle hundreds of pages of inputs ($>100k$ input tokens). As such, we do not think this provides a technical barrier to adversarial anonymization. Additionally, it has also previously been shown [5] that, e.g., GPT-4 outperforms existing methods in detecting PII in (longer) Human Right Court Documents [3].
> >
> > For millisecond applications, we agree with the reviewer that using a secondary model for anonymization may incur noticeable overhead. However, we are not targeting such particular instances nor did we encounter any such application in our literature research—if the reviewer has any particular application with such time complexity constraints in mind, we are interested to learn more about them. While specialized methods for such potential applications certainly would have their place (and arguably an additional different set of other requirements) we would argue that there is a wide majority of use-cases where ultra-low-latency is not a primary objective. Further, from an execution speed perspective, adversarial anonymization does not fare as badly as one may expect. From experience on our datasets, each text was anonymized in order of seconds per round (fully parallel between texts), which we deem very usable for most use-cases. Additionally, efficient and fast serving of LLMs is an active research and commercial application area, providing solutions with considerable inference time improvements over the methods employed for our evaluation.

---

> > > ### Author Response · Authors · 2024-11-22
> > >
> > > **Q3: How heavily dependent on LLM capabilities is adversarial anonymization? Can we end up in a constant race with new models and varying performance?**
> > >
> > > The reviewer raises a relevant point. We want to address this in two ways:
> > >
> > > With respect to the capabilities, we note that one can make the same argument about existing anonymization methods, in particular, as it has already been shown in our work and in [1, 5] that they are severely insufficient for actually anonymizing free-form text. As such, we are in a state where the adversary is strictly more capable than what current methods are offering. Crucially, overcoming this imbalance, adversarial anonymization offers a way to, *at minimum, achieve parity*. In particular, as we have shown empirically, smaller or finetuned (see Q4) already achieve very strong tradeoffs - even when compared to the strongest available models.
> > >
> > > This ties into our second argument, namely that these inferences are quite consistent across models and seem to transfer between models. Models like Llama3.1-70B and Llama3.1-8B perform so well exactly because they also anonymize the same underlying information as the stronger GPT-4. Additionally, for a given text, there is a finite amount of information that could be anonymized to prevent (reasonable) inferences as required by regulations. With many models already achieving close to human performance in this task, we do not expect an infinite race here; also, theoretically, in a given snippet of text there is an exhaustible amount of personal information contained. We agree with the reviewer that there could always be a stronger model in the future; however the closer we get to the human baseline and especially to the theoretical limit of private information inferable from a text, the less significant these improvements are from a privacy inference perspective. Further, our method even in its current instantiation already is a significant improvement over what span-based anonymization has to offer---providing a much better privacy-utility tradeoff. In addition, we can imagine adversarial anonymization to be particularly applicable in human-in-loop scenarios, where the human may recognize additional information they wish to hide but did not think of in the beginning.
> > >
> > > **Q4: Can you provide more insight into  failure cases of adversarial anonymization?**
> > >
> > > Absolutely. Prompted by the reviewer's question, we have manually checked all cases in the PersonalReddit dataset where the adversary was able to make a certain prediction ($\geq 4$) after the third iteration of GPT-4-AA. Unlike the cases with low certainty alluded to above, we consider these actual failure cases. Interestingly, we find that these failures (with the exception of a single case) are restricted to only three attributes: Education (5 cases), Relationship Status (7 cases), and Sex (10 cases). For all of these cases, we find that the core message of the text is closely related to the respective attribute, e.g., (for Sex) the personal experience of bearing a child or using gender-specific contraception (for Relationship Status) almost exclusively recent stories about experiences with dating (apps) that indicate their current relationship status as single and (for Education) mostly very college specific topics such as struggle with course load/selection. In all of the cases, a significant part of the utility of the text is given by the exposure of the personal attribute. While, e.g., more concrete references to universities have been removed from all texts, the overall nature of the text, which is about the life of a college student, was retained. In these cases, while adversarial anonymization provides some level of protection and certainly awareness for the user, the communication of the private attribute is core to the utility of the text, making full anonymization impossible without sacrificing almost all utility.
> > > We have included an additional discussion on this to the updated manuscript in App. G.

---

> > > > ### Author Response · Authors · 2024-11-22
> > > >
> > > > **Q5: Can one potentially distill these capabilities into smaller models?**
> > > >
> > > > We thank the reviewer for this suggestion. Based on this, we have fine-tuned a 4-bit version of Llama-3.1-8B (requiring only 4GB of memory) as a new inference model (keeping the original 8-bit Llama-3.1-8B as a repair model). In particular, we only used synthetic demonstrations from Llama3.1-70B on SynthaPAI and our synthetic samples and evaluated the newly trained model on the separate full real-world dataset. We provide a full overview over the setting and the results in the newly added App. I.
> > > > As we show in Fig. 20, this actually leads to a significant improvement in the utility-privacy-tradeoff, with the resulting Mixed-AA model achieving results almost as strong as the much larger LLama-3.1-70B. Interestingly we observed that demonstrations from Llama3.1-70B resulted in a better finetuned model than demonstrations from GPT-4. We speculate that this is due to the similar alignment of the Llama3.1 models. Overall, we believe these results are very promising, and more powerful approaches for model distillation could result in edge-device runnable models that achieve very strong utility-privacy tradeoffs. That is a very promising direction for future work, especially in order to enable lower-cost anonymization.
> > > >
> > > > **References**\
> > > > [1]  Staab, Robin, et al. "Beyond memorization: Violating privacy via inference with large language models." ICLR 2024.\
> > > > [2] Thomas Brewster. Chatgpt has been turned into a social media surveillance assistant, Nov 2023.\
> > > > [3] Pilán, I., et al. 2022, 12. “The Text Anonymization Benchmark (TAB): A Dedicated Corpus and Evaluation Framework for Text Anonymization.”\
> > > > [4] Lison, Pierre, et al. "Anonymisation models for text data: State of the art, challenges and future directions." Proceedings of the 59th Annual Meeting of the Association for Computational Linguistics and the 11th International Joint Conference on Natural Language Processing (Volume 1: Long Papers). 2021\
> > > > [5] Bubeck, Sébastien, et al. "Sparks of artificial general intelligence: Early experiments with gpt-4." arXiv preprint arXiv:2303.12712 (2023).

---

> > > > > ### Comment · Reviewer_18jd · 2024-11-25
> > > > >
> > > > > Thank authors for the rebuttal. The responses to Q4 and Q5 have effectively addressed my concerns, and I suggest that you include these experiments and analyses in the updated version. However, I still have reservations about the technical contributions and the fact that privacy can only be evaluated empirically due to the lack of theoretical analysis in this paper. This is not because your rebuttal is inadequate, I understand that you believe this is a new application and that you have conducted extensive experiments. Indeed, after reviewing your responses to Q4 and Q5, I feel the paper has improved. Nevertheless, my current impression is that this paper is still borderline work. Since there is no option for a score of 5.5, I will keep my original score.

---

> > > > > > ### Author Response · Authors · 2024-11-26
> > > > > >
> > > > > > We thank the reviewer for their feedback and are happy to hear that we could resolve multiple of their questions---we would also like to note that we uploaded a revision of the paper with the original rebuttal, including all referenced elements of our rebuttals to all reviewers. We appreciate the reviewer's feedback, which helped improve our paper and even pointed to new directions for future work.
> > > > > >
> > > > > > While we agree with the reviewer that a method providing theoretical guarantees would be the silver bullet for text anonymization, we argue that, to our knowledge, over a long line of anonymization research in NLP, **no method** was proposed that could provide **practical privacy guarantees** for the information leakage free-form text. Notably the current industry standard even falls significantly behind what can even be inferred with a 7B language model—highlighting that current anonymization does not even provide an empirical baseline related to the actual information leakage from text. For guarantees, we want to refer to a well-regarded paper in text anonymization [6] (similar arguments in [7]). In particular, methods from Privacy-Preserving Data Publishing (PPDP) that come with guarantees for structured databases (e.g., k-anonymity) commonly focus on the release of a full dataset, whereas  "[PPDP] solutions for anonymising unstructured text are scarce and mostly theoretical." Notably, DP methods (the gold standard for ML privacy) are generally considered in-applicable to individual text anonymizations [6].
> > > > > >
> > > > > > With this in mind, our work addresses a concrete issue in text anonymization, as pointed out in [6]: "NLP approaches to anonymization suffer from a number of shortcomings. Most importantly, they are **limited to predefined categories of entities** and **ignore how less conspicuous text elements may also play a role in re-identifying the individual.**" Crucially, it is primarily focused on detecting these predefined entities directly and exactly in a given text (ignoring most inferences). Our adversarial anonymization setting actively addresses this shortcoming by:
> > > > > >
> > > > > > Introducing a notion of anonymization directly related to the inferable information from text.
> > > > > > Providing an instantiation of an adversary that makes use of such less conspicuous text elements. Note that without such an adversary one has to rely on manually annotated datasets in order to evaluate any form of text-based privacy.
> > > > > > Showing in the real world how current defenses fall short against this practical adversary.
> > > > > > Providing an extensively evaluated approach for anonymization that achieves higher utility (including a human evaluation which is the best one can do for free-text utility)  and higher privacy protection.
> > > > > >
> > > > > > We believe there is a lot of scientific value in empirically quantifying and quite exhaustively setting the stage for much-improved text anonymization methods. Notably, we find that the setting we target (automated inferences from online texts) is something that is widely discussed academically [8][9][10][11][12] and in the privacy space [13][14][15] especially as it finds first applications in practice [16][17][18]. This gives practical relevance to the work presented here, which not only improves the setting but also provides first steps to protect users, which was not possible [8] and generally not studied before.

---

> > > > > > > ### Author Response · Authors · 2024-11-26
> > > > > > >
> > > > > > > **References**\
> > > > > > > [6] Lison, Pierre, et al. "Anonymisation models for text data: State of the  art, challenges and future directions." Proceedings of the 59th Annual  Meeting of the Association for Computational Linguistics and the 11th  International Joint Conference on Natural Language Processing (Volume 1: Long Papers). 2021\
> > > > > > > [7] Manzanares-Salor, B., Sánchez, D. & Lison, P. Evaluating the disclosure risk of anonymized documents via a machine learning-based re-identification attack. Data Min Knowl Disc 38, 4040–4075 (2024). https://doi.org/10.1007/s10618-024-01066-3 \
> > > > > > > [8] Robin Staab, Mark Vero, Mislav Balunovic, and Martin Vechev. Beyond memorization: Violating privacy via inference with large language models, ICLR, 2023\
> > > > > > > [9] Yukhymenko, Hanna, et al. "A Synthetic Dataset for Personal Attribute Inference." NeurIPS Dataset and Benchmarks, 2024.\
> > > > > > > [10] Dou, Yao, et al. "Reducing Privacy Risks in Online Self-Disclosures with Language Models." arXiv preprint arXiv:2311.09538 (2023).\
> > > > > > > [11] Yao, Yifan, et al. "A survey on large language model (llm) security and privacy: The good, the bad, and the ugly." High-Confidence Computing (2024): 100211. \
> > > > > > > [12] Li, Haoran, et al. "Privacy in large language models: Attacks, defenses and future directions." arXiv preprint arXiv:2310.10383 (2023).\
> > > > > > > [13] Schneier, Bruce. “The Internet Enabled Mass Surveillance. A.I. Will Enable Mass Spying.” Slate, 4 Dec. 2023, slate.com/technology/2023/12/ai-mass-spying-internet-surveillance.html. \
> > > > > > > [14] Moody, Glyn. “ChatGPT Is One Year Old: Here’s AI’s next Attack on Privacy, and What to Do about It.” Private Internet Access Blog, 8 Dec. 2023, www.privateinternetaccess.com/blog/chatgpt-is-one-year-old-heres-ais-next-attack-on-privacy-and-what-to-do-about-it/. Accessed 26 Nov. 2024.\
> > > > > > > [15] Stanley, Jay. “Will ChatGPT Revolutionize Surveillance? | ACLU.” American Civil Liberties Union, 19 Apr. 2023, www.aclu.org/news/privacy-technology/will-chatgpt-revolutionize-surveillance. \
> > > > > > > [16] Brewster, Thomas. “ChatGPT Has Been Turned into a Social Media Surveillance Assistant.” Forbes, 16 Nov. 2023, www.forbes.com/sites/thomasbrewster/2023/11/16/chatgpt-becomes-a-social-media-spy-assistant/?sh=56b7f0345cf6. \
> > > > > > > [17] Levinson-Waldman, Rachel, et al. “Social Media Surveillance by the U.S. Government.” Brennan Center for Justice, 7 Jan. 2022, www.brennancenter.org/our-work/research-reports/social-media-surveillance-us-government. \
> > > > > > > [18] SOCIAL LINKS. “Social Links - OSINT Tools for Investigations.” Sociallinks.io, sociallinks.io/. \

---

### Official Review · Reviewer_EPkh · 2024-11-01

**Soundness:** 4
**Presentation:** 4
**Contribution:** 2
**Rating:** 5
**Confidence:** 5

**Summary:**

In this work, the authors focus on the privacy scenario where online texts can be exploited to infer personal data. The authors utilize adversarial LLM inferences, which are highly performant in extracting personal attributes from unprotected texts, for evaluating anonymization and also use this adversarial model as "feedback provider" to another LLM whose goal is to anonymize texts. An iterative framework between these two LLMs lead to strong anonymization performance as shown by the authors in wide range of experiments, outperforming existing anonymizers and also aligning well with human preference.

**Strengths:**

The presentation is very clear and the paper flows very well. Text anonymization is an important problem in the realm of privacy and the approach the authors introduce do improve the existing anonymization tools significantly. The evaluation section has extensive analysis, which is great. The reviewer really enjoyed reading this paper overall.

**Weaknesses:**

In my opinion, the paper is very well written and the authors conducted extensive empirical studies to demonstrate the significant improvement of their approach compared to the existing text anonymization tools. I think my main concern is the scope and complexity of the approach appear quite limited, especially for a conference of this caliber. The approach is based on iterating two LLMs, one is anonymizing text and the other is trying to infer personal attributes. To me, this is like a cute application of LLMs but perhaps rather better suited for a workshop instead of this conference. In this sense, I am unsure about the fit.

**Questions:**

1. Have you considered measuring utility by some downstream applications? E.g. if the texts are used for some analysis or for some task, how the performance changes from the original unprotected texts to the anonymized texts. Would you think this could also serve for useful utility metrics?

2. How can one turn this approach into a more comprehensive privacy-protecting tool? To my understanding, it currently builds on pre-defined set of attributes and the adversary LLM is trying to infer these attributes while the anonymizer LLM is trying to anonymize as oppose.  But it'd be hard to list all possible attributes that could lead to deducing personal information so any comments on scaling this approach would be appreciated.

3. Also related to my question above, formal privacy guaranteeing mechanisms like differential privacy ensures that even the existence of the data cannot be inferred from the analysis by any adversary. Although in this work the authors focus on anonymizing individual text snippets so that DP may not be applicable, however, it'd be interesting to find a common scenario where two approaches can be compared I think.

Minor: AzureLanguageService -> Azure Language Service

---

> ### Author Response · Authors · 2024-11-22
>
> We thank the reviewer for their comprehensive feedback. We respond to the raised concerns below, numbered Q1-Q4. We have also uploaded a revised version of the paper, with updates highlighted in violet. Further we thank the reviewer for catching typos which we adapted in the manuscript. We are happy to continue the discussion further if the reviewer has additional questions.
>
>
> **Q1: Does the simplicity of the final anonymization algorithm decrease the overall merit of the work?**
>
> No, we believe that even though our eventual anonymization method is conceptually straightforward from a purely technical perspective, it makes significant contributions in the field of text anonymization in several ways—especially so in casual communicative settings, such as online forums, where, as shown in [1] LLMs pose a severe novel threat to anonymity.
>
> On the setting side, we are the first to provide an (inference-based) adversarial view on text anonymization, addressing the key limitation of prior stance on anonymization; namely that even entity-blanking-based anonymization still leaks private information in presence of an inferential adversary [1]. This is a critical issue that is fundamental to current anonymization tools and evaluation. As also shown in [1], we demonstrate in our evaluation that on real-world data existing methods are clearly insufficient both against adversaries and in utility.
>
> On a practical side, as reviewers agree, adversarial-feedback anonymization using LLMs provides a natural instantiation of this setting and achieves consistently better privacy protection and utility than *commercial* anonymization tools. As such, our strong empirical contributions could have an noticable impact on the state of anonymization currently in application.
>
> Further, having such an elaborate analysis and extensive experimental evaluation (across 17 models, multiple datasets, and including a human study), sets a strong baseline for any potential follow-up work in this area.
>
> As such, we believe that there is significant merit in (1) defining a practical and free form setting for text anonymization and (2) providing a method that outperforms current industry-standard anonymization across very extensive evaluations. Such a contribution is particularly timely as developments in LLMs have made such inferences much easier [1] up to the point where it can be potentially misused on a large scale [2], constituting a practical real-world threat.

---

> ### Author Response · Authors · 2024-11-22
>
> **Q2: Have you considered measuring the utility via downstream performance? For which settings do you believe adversarial anonymization is useful here?**
>
> One of the key challenges with free-form text in our target online setting is that utility in many cases (particularly for users that write the texts) is defined via the coherence and expression of the text itself. Notably measuring the utility via some downstream task can actually hide a lot of utility-loss (as we detail further in Q4) that would be important to humans in this setting (such as readability). Nevertheless, if there is a clear downstream target one could also use this to get a practical indication of how much task-specific utility is retained. Prompted by the reviewer's question, we extended our evaluation in two ways: First, we evaluate also on the MedQA dataset [3] having a much more rigid text setting, with a clearly defined downstream MC-Accuracy metric, and secondly, we compute embeddings over all anonymized (and original) texts in PersonalReddit allowing us to quantify their similarity independent of potential downstream tasks.
>
> On MedQA using GPT-4o as a downstream classifier we achieve $85.4\%$ baseline accuracy. After applying a single round of adversarial anonymization we reduce the number of adversarial age, location, and place-of-birth, predictions by $>50\%$ while also showing strong results other attributes like Sex ($>25\%$) and occupation, while still maintaining a downstream accuracy of $81.4\%$ (we expect some drop as in some cases this information can be quite relevant for predictions). This makes it competitive in utility with Azure ($81.5\%$) that works that works better on such reports than on free-form data (e.g., almost every text starts with “A XX-years old man/woman/baby has been …”), while slightly outperforming it on privacy. We present a full overview in App. H.1.
>
> As a proxy for the retention of downstream utility on free-form text, we further also compute embeddings using the \texttt{text-embedding-3-large} model by OpenAI. As these embeddings are usable for all sorts of potential downstream tasks, they constitute a strong proxy for how well we might perform on arbitrary downstream tasks and quantify how close the anonymized text is to the original. We report cosine similarity between embedding vectors (1 being a perfect match). We find that, e.g., on Llama-3.1-70B-AA, we have a median cosine-similarity of $0.93$ after one round ($0.88$ after 2 and $0.84$ after 3). This is in stark contrast to the median of $0.24$ between $1000$ selected random comments in PersonalReddit and re-affirms that adversarial anonymization maintains a high level of utility.
>
> We provide more details and an additional discussion on the above experiments in the newly added App. H.
>
> **Q3: How can this be turned into a more comprehensive tool? Does adversarial anonymization require me to pre-define all attributes that should be removed in the text? Is this feasible?**
>
> The reviewer raises an interesting question. In particular, we find that Adversarial anonymization only requires one to define the attributes that they want to protect, not the way they are expressed in text. As such, there could be no unintended privacy leakage in case the user lists everything to the algorithm that they consider “private”. In fact, these “attributes” may be even more complex aspects to protect, e.g., mental health information or descriptions of any specific information to hide. This is a natural improvement over how it is done in classical anonymizers (Presidio, Azure) that target the direct expression of a clearly scoped attribute in the text (e.g., a direct mention of a location but not clues that would allow a very certain inference). The interpretation of adversarial anonymization is much more aligned with regulatory requirements here (e.g., [4]). However, as pointed out by the reviewer, this still requires us to define the initial set of attributes. In a pragmatic sense, these can quite often be directly informed by existing legislation ([4] and [5]). Note that theoretically, as a first step, the adversarial LLM could be tasked to “infer everything and anything” from the given text, from which then the user could select which information they wish to remove from the snippet. The deeper investigation of the promise of such approaches is, we believe, an interesting avenue for future work.

---

> ### Author Response · Authors · 2024-11-22
>
> **Q4: Can you relate adversarial anonymization to DP? In which cases could there be joint applications?**
>
> We agree with the reviewer that DP's privacy guarantees are the gold standard in many settings. However, in practice, these mechanisms are often very difficult to apply to text (For instance as in [6] (When did Tesla move to New York City? -> Wave did Tesla It way Dru Tully breaking?)), especially when we want to keep the general utility and meaning of the text, such as in free form online settings considered in our work. This problem is exacerbated in the setting we are targeting, where people are individually contributing to a "database" and have to resort to local DP approaches. Further, potential future interactions of a user are unbounded, inhibiting rigorous and finite DP guarantees that hold also in the future. Finally, DP fundamentally requires randomness, which would mean that DP-based anonymization would not allow the user to interfere with the anonymized outputs. Our previous example illustrates well why this is a problem; imagine if a user wants to ask “When did Tesla move to New York City?” in a forum but once they hit the “post” button, the posted text appears as “Wave did Tesla It way Dru Tully breaking?”, without an option to edit. In contrast, our method enables more practical fine-grained interactions with the user at each round of anonymization.
>
> Nonetheless, we believe that adversarial anonymization and DP can actually complement each other as they target different notions of privacy - in particular, we can imagine a setting in which the collected data text is first anonymized in order to protect individual attributes at the time of collection and, later, model training via DP-SGD provides privacy guarantees on that particular model instance itself. If the reviewer is interested in a more comprehensive overview of text anonymization and DP methods we can recommend [7] as a great starting ressource.
>
> **References**\
> [1]  Staab, Robin, et al. "Beyond memorization: Violating privacy via inference with large language models." ICLR 2024.\
> [2] Thomas Brewster. Chatgpt has been turned into a social media surveillance assistant, Nov 2023.\
> [3] Jin, Di, et al. "What disease does this patient have? a large-scale open domain question answering dataset from medical exams." Applied Sciences 11.14 (2021): 6421.\
> [4] DOL, 2023. URL https://www.dol.gov/general/ppii. \
> [5] European Union. (n.d.). What personal data is considered sensitive?. European Commission.https://commission.europa.eu/law/law-topic/data-protection/reform/rules-business-and-organisations/legal-grounds-processing-data/sensitive-data/what-personal-data-considered-sensitive_en. \
> [6] Yue, Xiang, et al. "Differential privacy for text analytics via natural text sanitization." arXiv preprint arXiv:2106.01221 (2021).\
> [7] Lison, Pierre, et al. "Anonymisation models for text data: State of the art, challenges and future directions." Proceedings of the 59th Annual Meeting of the Association for Computational Linguistics and the 11th International Joint Conference on Natural Language Processing (Volume 1: Long Papers). 2021.

---

> > ### Comment · Reviewer_EPkh · 2024-11-25
> > **Response to the authors**
> >
> > The reviewer appreciates the detailed discussion provided by the authors and also the additional empirical studies. The reviewer does maintain their concern about the fit but will definitely consider increasing their score.

---

> > > ### Author Response · Authors · 2024-11-26
> > >
> > > We thank the reviewer for their consideration and for providing feedback that resulted in an improved version of the manuscript—we truly appreciate it. Given the significant improvements over industry-level anonymization, level of exposition, and evaluation presented in the paper (17 models, 4 datasets, human study, various ablations, finetuning outlook), we believe the work is fit for a venue such as ICLR and establishes a strong foundation for future work in this area. If it helps assure the reviewer about the overall fit, we are happy to find that there are already works not affiliated with us (under submission in this cycle at ICLR) that build on / compare to our work.
> > >
> > > But mostly, we believe that in light of the results of [1], showing that current text anonymization is ineffective in the presence of LLMs and current anonymization evaluation is disconnected from practical privacy risks in text, it is evident that fundamental adjustments have to be made to current tools and practices in anonymization. In this regard, we view our work as significant as it, for the first time, introduces a view on text anonymization that is practically linked to the inferential privacy leakage from the text and proposes an effective method to utilize the strongest available adversary for text sanitization. While we agree (and hope) that there is room for more complex methods and theoretical analyses, we believe that our work can lay the solid empirical and conceptual foundation for these important future studies.

---

### Official Review · Reviewer_CnVX · 2024-11-04

**Soundness:** 3
**Presentation:** 3
**Contribution:** 3
**Rating:** 8
**Confidence:** 3

**Summary:**

This paper introduces a novel approach to text anonymization in the era of large language models. The authors present two main contributions: (1) a new evaluation framework that leverages LLMs for adversarial inference to measure anonymization effectiveness, and (2) an iterative anonymization pipeline that uses adversarial feedback to guide the text anonymization process. This framework offers improvement over the traditional span based formulation as contextual information leaks information as well. The authors conduct extensive experiments with various models and demonstrate that their approach achieves better privacy-utility tradeoffs compared to traditional span-based anonymization techniques such Azure Language Services. In their results, performing the procedure reduces the adversarial inference chance from 87% to 66%, and iterating the procedure with GPT-4 for three rounds further reduces adversarial inference success to ~45% while maintaining higher text utility than baseline methods. They validate their results with human annotation.

**Strengths:**

- Novel approach that leverages LLMs' inference capabilities to measure privacy leakage in a more realistic way than traditional span-based methods.
- Comprehensive experimental evaluation across multiple models, attributes, and metrics, with clear ablation studies showing the benefit of the feedback-guided approach.
- Strong empirical results showing significant improvements over industry-standard tools like Azure Language Service, with detailed analysis of both privacy protection and utility preservation.
- Thoughtful consideration of practical concerns including computational costs, local deployment options, and regulatory compliance.
- Clear demonstration of how multiple rounds of anonymization can progressively improve privacy while maintaining readable text.

**Weaknesses:**

- The ~41% remaining adversarial inference success rate after anonymization remains concerning for privacy-critical applications. The paper would benefit from deeper analysis of these failure cases.
- Limited domain evaluation, focusing primarily on data directly or indirectly from Reddit. Testing on other domains (medical, legal, etc.) would strengthen generalizability claims. It could also be reddit comments that mentions information from these domains.
- The method requires pre-defining attributes to protect, which may miss unexpected privacy leaks. An automated approach to identifying sensitive attributes could be valuable.
- Cost analysis could be more comprehensive - while per-comment costs are reasonable (
(~$0.035), real-world applications with high volume could face significant expenses. In addition, the estimate is only for one turn, so to achieve the same level of privacy protection, this number might be more expensive.

**Questions:**

- Have you explored methods to automatically determine the optimal number of iterations, perhaps based on inference confidence?
- How does the system perform when encountering privacy-sensitive attributes not explicitly listed in the input? Could it be extended to automatically identify such attributes?
- Can you provide specific examples of common failure cases where privacy leaks persist even after multiple rounds of anonymization? Understanding these patterns could help improve the approach.
- Have you considered how this approach might need to be adapted for different domains with varying privacy requirements and linguistic patterns?

---

> ### Author Response · Authors · 2024-11-22
>
> We thank the reviewer for their comprehensive feedback. We respond to the raised concerns below, numbered Q1-Q7. We have also uploaded a revised version of the paper, with updates highlighted in violet. We are happy to continue the discussion further if the reviewer has additional questions.
>
> **Q1: Does the remaining 41% accuracy after anonymization indicate failures of the anonymization?**
>
> We thank the reviewer for raising this important point. To clarify, no, it does not indicate the failure of anonymization, it is instead an artifact of highly uncertain predictions in later anonymization stages. First, we note that certain discrete features enable even a random adversary to achieve high-looking scores, e.g., random accuracy on sex would be ~50%. Further, to elaborate on the main reason behind the remaining accuray, note that in the reported joint adversarial accuracy of 41%, we count any prediction made by the adversary for any certainty level (1-5). In particular, this also accounts for cases when there was no clear evidence or basis for inference in the text and the model relied purely on inherent biases (i.e., certainty 1 and 2 - we provide qualitative examples in the newly added App. D.4).
>
> Crucially, if we only account for predictions made with certainty $3$ or higher (following [1]) the resulting adversarial accuracy drops down to only $7.7%$ (For $\geq 2$ we get $19\%$. Notably as we had shown in Fig. 4 this even furthers the advantage adversarial anonymization has over traditional techniques as it not only leads to lower accuracy in guessing but also a much lower certainty in the guesses.
>
> Prompted by the reviewers question we have added an additional discussion on this to the updated manuscript in App. G. We are happy to expand on this in case the reviewer has further recommendations.
>
> **Q2: Why did you evaluate particularly on the online domain setting? How does adversarial anonymization transfer to other settings?**
>
> We primarily focused on the online domain as (1) it is a setting where real-world risks were shown in prior works [1,2] and (2) it is a setting which is uniquely challenging for existing methods in anonymization—to an extent where they are borderline unusable (and unpreferred, as shown in our human evaluation). In that sense we believe that providing a solution that works well here is a valuable contribution.
>
> Prompted by the reviewer's question, we extended our evaluation in two ways (presenting new results in App. H): First, we evaluate also on the MedQA dataset [2] having a much more rigid/structured text setting, with a clearly defined downstream utility metric via multiple-choice accuracy. Second, we compute embeddings over all anonymized (and original) texts in PersonalReddit allowing us to quantify their similarity independent of potential downstream tasks.
>
> On MedQA using GPT-4o as a downstream classifier we achieve $85.4\%$ baseline accuracy. After applying a single round of adversarial anonymization we reduce the number of adversarial age, location, and place-of-birth, predictions by $>50\%$ while also showing strong results other attributes like Sex ($>25\%$) and occupation, while still maintaining a downstream accuracy of $81.4\%$ (we expect some drop as in some cases this information can be quite relevant for predictions). This makes it competitive in utility with Azure ($81.5\%$) that works that works better on such reports than on free-form data (e.g., almost every text starts with “A XX-years old man/woman/baby has been …”), while slightly outperforming it on privacy. We present a full overview in App. H.1.
> As a proxy for the retention of downstream utility on free-form text, we further also compute embeddings using the \texttt{text-embedding-3-large} model by OpenAI. As these embeddings are usable for all sorts of potential downstream tasks, they constitute a strong proxy for how well we might perform on arbitrary downstream tasks and quantify how close the anonymized text is to the original. We report cosine similarity between embedding vectors (1 being a perfect match). We find that, e.g., on Llama-3.1-70B-AA, we have a median cosine-similarity of $0.93$ after one round ($0.88$ after 2 and $0.84$ after 3). This is in stark contrast to the median of $0.24$ between $1000$ selected random comments in PersonalReddit and re-affirms that adversarial anonymization maintains a high level of utility.
> We provide more details and an additional discussion on the above experiments in the newly added App. H.

---

> > ### Author Response · Authors · 2024-11-22
> >
> > **Q3: Does adversarial anonymization require me to pre-define all attributes that should be removed in the text? Is this feasible?**
> >
> > The reviewer raises an interesting question. In particular, we find that adversarial anonymization only requires one to define the attributes that they want to protect, not the way they are expressed in text. As such, there could be no unintended privacy leakage in case the user lists everything to the algorithm that they consider “private”. In fact, these “attributes” may be even more complex aspects to protect, e.g., mental health information or descriptions of any specific information to hide. This is a natural improvement over how it is done in classical anonymizers (Presidio, Azure) that target the direct expression of a clearly scoped attribute in the text (e.g., a direct mention of a location but not clues that would allow a very certain inference). The interpretation of adversarial anonymization is much more aligned with regulatory requirements here [3]. However, as pointed out by the reviewer, this still requires us to define the initial set of attributes. In a pragmatic sense, these can quite often be directly informed by existing legislation ([3] and [4]), which is why we selected this as an (extensive) baseline. Note that theoretically, as a first step, the adversarial LLM could be tasked to “infer everything and anything” from the given text, from which then the user could select which information they wish to remove from the snippet. The deeper investigation of the promise of such approaches is, we believe, an interesting avenue for future work.
> >
> > **Q4: Can you elaborate on the cost calculations of adversarial anonymization?**
> >
> > Certainly, and we have also extended App. A.6 to include a more detailed discussion in our updated manuscript. First and foremost, we agree with the reviewer that adversarial anonymization is more costly than, e.g., running regexes over an input string. Our currently presented number estimates this cost more on the worst-case side. In particular, since then, the cost of running the latest GPT-4 has reduced by 4x in input and 3x output (even 8x and 6x - when running in batched mode). The cost is further reduced if we switch to open models (or, as we show in the newly added App. I - are able to distill knowledge into a smaller model) - with the cost for Llama-3.1-8B-AA being only $\$0.002$ for the full five rounds. While this cost, of course, is still higher than doing basic traditional NLP, we believe that there are many potential applications (especially for smaller local models) where this increase in cost is justified by the increase in both utility and privacy. In these cases, adversarial anonymization is both a valuable contribution and a (feasible) tool. Also note that presumably in the near future small LLMs may be integrated into mobile devices, which could be then used to power an anonymization application based on our method at almost no marginal cost.
> >
> > **Q5: Have you explored methods to automatically determine a cutoff point for the number of iterations?**
> >
> > The reviewer raises an interesting point. In particular, in our newly added App. G, we include an ablation over the relative accuracy of the adversary based on the certainty that it presents (overall iterations of GPT-4-AA). This plot strongly indicates that when the certainty in a prediction falls below $\leq 2$, there is only a marginal benefit in running additional iterations. Based on our runs on real-world data, we find that this already quite often happens after 1 or 2 rounds (see new Figure 17, $~50\%$ of predictions have certainty $\leq 2$ after a single round), allowing in many of these cases to stop early (not only resulting in lower costs but also higher utility). Particularly helpful is here, in practice, the fact that we, as the anonymizing party, can often assume knowledge of the actual attribute values. As such, we can actually directly check against the current inference to test whether an attribute is inferable (a generally stronger criterion).

---

> > > ### Author Response · Authors · 2024-11-22
> > >
> > > **Q6: Can you provide common failure cases for adversarial anonymization?**
> > >
> > > Absolutely. Prompted by the reviewer's question, we have manually checked all cases in the PersonalReddit dataset where the adversary was able to make a certain prediction ($\geq 4$) after the third iteration of GPT-4-AA. Unlike the cases with low certainty alluded to above, we consider these actual failure cases. Interestingly, we find that these failures (with the exception of a single case) are restricted to only three attributes: Education (5 cases), Relationship Status (7 cases), and Sex (10 cases). For all of these cases, we find that the core message of the text is closely related to the respective attribute, e.g., (for Sex) the personal experience of bearing a child or using gender-specific contraception (for Relationship Status) almost exclusively recent stories about experiences with dating (apps) that indicate their current relationship status as single and (for Education) mostly very college specific topics such as struggle with course load/selection. In all of the cases, a significant part of the utility of the text is given by the exposure of the personal attribute. While, e.g., more concrete references to universities have been removed from all texts, the overall nature of the text, which is about the life of a college student, was retained. In these cases, while adversarial anonymization provides some level of protection and certainly awareness for the user, the communication of the private attribute is core to the utility of the text, making full anonymization impossible without sacrificing almost all utility.
> > >
> > > **Q7: Have you considered possible adaptations for other privacy requirements and linguistic patterns?**
> > >
> > > This is a very interesting avenue. For now, we have focussed our setup primarily on English and the attributes defined in existing prominent privacy regulations. With this in mind, the framework is straightforward enough that we have seen it work directly in other languages (e.g., our dataset contains some comments (partially) in Spanish, French, and German). Especially with the development of more language-specific LLMs (as well as the growing multilingual capabilities of frontier models), adversarial anonymization is very likely to produce similarly strong results here. With respect to varying privacy settings, this becomes even more interesting. We already observed a stronger degree of application flexibility in AA than in traditional text anonymizers (as well as strong results across several domains). Further, there is actually even one work currently under review at ICLR that uses AA as a baseline for a linkability privacy setting. If the reviewer has particular settings in mind here, we would be very excited to hear about them!
> > >
> > > **References**\
> > > [1]  Staab, Robin, et al. "Beyond memorization: Violating privacy via inference with large language models." ICLR 2024.\
> > > [2] Jin, Di, et al. "What disease does this patient have? a large-scale open domain question answering dataset from medical exams." Applied Sciences 11.14 (2021): 6421.\
> > > [3] DOL, 2023. URL https://www.dol.gov/general/ppii. \
> > > [4] European Union. (n.d.). What personal data is considered sensitive?. European Commission.https://commission.europa.eu/law/law-topic/data-protection/reform/rules-business-and-organisations/legal-grounds-processing-data/sensitive-data/what-personal-data-considered-sensitive_en

---

> ### Comment · Reviewer_CnVX · 2024-12-03
> **Rebuttal Acknowledgement**
>
> I appreciate the authors' rebuttal, which has sufficiently addressed my concerns. I have increased my score as a result. After reading all the other discussion, I note that the primary criticism centers on the method's lack of novelty. While it is less satisfying that this work does not provide theoretical guarantees or propose a novel method to protect privacy, it presents a fresh and more practical perspective on privacy that is backed up with extensive experiments.

---

> > ### Author Response · Authors · 2024-12-03
> >
> > We are glad that our rebuttal addressed the reviewer's concerns, and we thank the reviewer for raising their score and for providing constructive feedback that has led to a better version of the paper. We are happy to hear that the reviewer shares the view that our work presents a fresh and more practical perspective on privacy backed up by extensive experiments.

---

### Author Response · Authors · 2024-11-22
**General response**

We thank the reviewers for their feedback and their thorough evaluation of our work.
We are pleased that reviewers find that our work introduces a more realistic and practical setting for privacy measurement (CnVX), provides a method that shows noticeably empirical improvements over existing solutions (CnVX, EPkh, 18jd) and has an extensive evaluation (CnVX, EPkh) including a human study (18jd).
We are also glad to hear that reviewers found the paper clear and thoughtful with respect to practical considerations (CnVX) as well as enjoyable to read (EPkh).

Alongside this rebuttal we have uploaded an updated version of our manuscript (with new content in purple), that contains various additional results and explanations to support our rebuttal. We responded to each reviewer separately below and are happy to further engage in the discussion in case of follow-up questions.

---

### Meta-Review · Area_Chair_QLQe · 2024-12-22

**Metareview:**

Building on prior work from Staab et al. (2023) that showed that LLMs can infer many personal attributes from online posts without needing explicit identifiers, the authors show that LLMs can still infer these attributes even after the posts have been "anonymized" by SOTA, industry-grade anonymizers. They then propose an iterative adversarial scheme to better anonymize these posts (essentially, prompting an LLM to see if they can still infer any personal attribute; and then editing the post accordingly).

Reviewers thought the paper was well-written, well-executed, and tackling an important problem. The main concern from some reviewers was about fit, given that the paper is largely prompting-based and does not introduce any other methodological or conceptual developments. I am less concerned about this issue because the paper provides an important contribution by showing that SOTA anonymizers still allow for significant inference of personal attributes and that simple iterative methods with an LLM in the loop can improve anonymization. While there are no "novel technical contributions", I think this paper carries a useful message for the community and can serve as a foundation for future efforts to build better privacy safeguards. Thus, I recommend acceptance. I encourage the authors to take the skeptical reviewer feedback into account in the framing of their paper.

**Additional Comments On Reviewer Discussion:**

Not much discussion besides clarifications; most of the objections are philosophical about what degree of technical contribution is needed for a good paper.

---

### Decision · Program_Chairs · 2025-01-22

Accept (Poster)